# Dynamic changes in anti-SARS-CoV-2 antibodies during SARS-CoV-2 infection and recovery from COVID-19

Kening Li[1,2,3,13], Bin Huang[2,3,13], Min Wu[2,3,13], Aifang Zhong[4,5,13], Lu Li[2,3], Yun Cai[2,3], Zhihua Wang[5,6], Lingxiang Wu[2,3], Mengyan Zhu[2,3], Jie Li[2,3], Ziyu Wang[2,3], Wei Wu[2,3], Wanlin Li[2,3], Bakwatanisa Bosco [2,3], Zhenhua Gan[7,8], Qinghua Qiao[5,9], Jian Wu[10], Qianghu Wang [1,2,3,11,14✉], Shukui Wang[12,14✉] & Xinyi Xia [5,8,10,14✉]

Deciphering the dynamic changes in antibodies against SARS-CoV-2 is essential for understanding the immune response in COVID-19 patients. Here we analyze the laboratory findings of 1,850 patients to describe the dynamic changes of the total antibody, spike protein (S)-, receptor-binding domain (RBD)-, and nucleoprotein (N)-specific immunoglobulin M (IgM) and G (IgG) levels during SARS-CoV-2 infection and recovery. The generation of S-, RBD-, and N-specific IgG occurs one week later in patients with severe/critical COVID-19 compared to patients with mild/moderate disease, while S- and RBD-specific IgG levels are 1.5-fold higher in severe/critical patients during hospitalization. The RBD-specific IgG levels are 4-fold higher in older patients than in younger patients during hospitalization. In addition, the S- and RBD-specific IgG levels are 2-fold higher in the recovered patients who are SARS-CoV-2 RNA negative than those who are RNA positive. Lower S-, RBD-, and N-specific IgG levels are associated with a lower lymphocyte percentage, higher neutrophil percentage, and a longer duration of viral shedding. Patients with low antibody levels on discharge might thereby have a high chance of being tested positive for SARS-CoV-2 RNA after recovery. Our study provides important information for COVID-19 diagnosis, treatment, and vaccine development.

[1] The Affiliated Cancer Hospital of Nanjing Medical University, Jiangsu Cancer Hospital, Jiangsu Institute of Cancer Research, Nanjing, China. [2] Center for Global Health, School of Public Health, Nanjing Medical University, 211166 Nanjing, Jiangsu, China. [3] Department of Bioinformatics, Nanjing Medical University, 211166 Nanjing, Jiangsu, China. [4] Medical Technical Support Division, the 904th Hospital, 213003 Changzhou, Jiangsu, China. [5] Department of Laboratory Medicine & Blood Transfusion, Wuhan Huoshenshan Hospital, 430100 Wuhan, Hubei, China. [6] Department of Laboratory Medicine & Blood Transfusion, the 907th Hospital, 350702 Nanping, Fujian, China. [7] Department of Medical Administration, Jinling Hospital, Nanjing University School of Medicine, 210002 Nanjing, Jiangsu, China. [8] Joint Expert Group for COVID-19, Wuhan Huoshenshan Hospital, 430100 Wuhan, Hubei, China. [9] Medical and Technical Support Department, Pingdingshan Medical District, the 989th Hospital of Joint Logistic Support Force, 467000 Pingdingshan, Henan, China. [10] COVID-19 Research Center, Institute of Laboratory Medicine, Jinling Hospital, Nanjing University School of Medicine, 210002 Nanjing, Jiangsu, China. [11] Collaborative Innovation Center for Cardiovascular Disease, Nanjing Medical University, 211166 Nanjing, China. [12] Department of Laboratory Medicine, Nanjing First Hospital, Nanjing Medical University, 210006 Nanjing, Jiangsu, China. [13] These authors contributed equally: Kening Li, Bin Huang, Min Wu, Aifang Zhong. [14] These authors jointly supervised this work: Qianghu Wang, Shukui Wang, Xinyi Xia. ✉email: wangqh@njmu.edu.cn; sk_wang@njmu.edu.cn; xiaxynju@163.com

Coronavirus disease 2019 (COVID-19), which is caused by severe acute respiratory syndrome coronavirus 2 (SARS-CoV-2) infection, is spreading in more than 210 countries and territories globally[1–3]. As of August 19, 2020, a total of 21,938,171 confirmed cases were reported, of which 775,581 patients died. The high infection rate of SARS-CoV-2 leads to its rapid spread[4]. Over 100,000 confirmed cases were reported daily, creating major challenges for public health and medical services around the world. Therefore, rapid diagnosis and specific therapy for COVID-19 are urgently needed.

Currently, the diagnosis of COVID-19 is mainly based on testing SARS-CoV-2 RNA load using quantitative real-time polymerase chain reaction (RT-PCR)[5]. However, the nucleic acid testing results are subject to many factors, including the specimen site, type, quality, and patients' condition, and sample storage. Thus, some individuals with COVID-19 will remain undiagnosed if the diagnosis is based solely on the viral RNA load[6]. In consideration of the high false-negative rate of viral RNA detection, on March 3, 2020, SARS-CoV-2-specific IgM and IgG antibody levels were added to the "Diagnosis and Treatment Protocol for Novel Coronavirus Pneumonia of China" as alternative methods to diagnose the suspected cases. Antibody detection is simpler and faster than viral RNA load testing, and the test samples are more stable and easier to store[7]. Thus, antibody tests can provide an important complementary method for the diagnosis of COVID-19.

In addition, the generation and maintenance of neutralizing antibodies against SARS-CoV-2 play an important role in resisting infection by host[8]. SARS-CoV-2 belongs to the *Betacoronavirus* genus in the family *Coronaviridae*, which includes four primary proteins: spike (S), envelope (E), membrane (M), and nucleocapsid (N). The S protein is composed of S1 and S2 subunits, and S1 is responsible for the binding between the virus and host cell receptors. There is a receptor-binding domain (RBD) in the S1 subunit, which interacts with human cells that express angiotensin-converting enzyme 2 (ACE2) and induces entry of the virus[8]. Neutralizing antibodies often target the RBD of the S protein to block the interaction between the virus and the host receptor[9]. Antibodies against S protein, especially the RBD of SARS-CoV, serve as a target for the development of vaccines and therapy[10]. Recent studies have reported significant progress in the development of COVID-19 therapy and vaccines based on the S protein or RBD[11–14]. Chi et al.[11] isolated and characterized a neutralizing monoclonal antibody binds to the S protein of SARS-CoV-2 from ten convalescent COVID-19 patients. Dai et al.[14] reported a universal design of *Betacoronavirus* vaccines against COVID-19, Middle East respiratory syndrome, and severe acute respiratory syndrome based on the RBD-dimer structure. Several studies demonstrated that the RBD-specific IgG titer and viral neutralization titer had a strong positive correlation[15–17]. There was no evidence that N-specific antibodies can block virus infection. However, the S-, RBD-, and N-specific antibody responses against SARS-CoV-2 during COVID-19 infection and recovery are still unclear, especially the differences among patients with different ages, severity, and outcome.

Here, we analyze the laboratory tests of 1850 hospitalized COVID-19 patients. We describe the dynamic changes of the SARS-CoV-2-specific antibody levels, including the total, S-, RBD-, and N-specific IgM and IgG levels on admission, during hospitalization, and on discharge, and the relationship between viral shedding and the antibody response.

SARS-Cov-2 infection, we analyzed the levels of total antibody, and the S-, RBD-, N-specific antibodies at different time points after symptom onset using two commercial kits (see "Methods"). The first kit was used to detect the total antibodies, including IgM/IgG against S or N proteins for sensitively diagnosing COVID-19. The other kit was used to detect the S-, RBD-, N-specific IgM/IgG for analyzing the underlying immune response process of COVID-19 patients. The laboratory test results of 1850 hospitalized COVID-19 patients were analyzed (Supplementary Table 1). For the detection of total IgM/IgG, 3058 serum samples from 1850 patients were tested. Each patient was tested one to ten times, and 669 (36.2%) were tested more than once. The median sampling interval was 5 days among patients who were tested more than once. For the detection of S-, RBD- and N-specific antibodies, 712 serum samples from 418 patients were tested. Each patient was tested one to seven times, and 169 (40.0%) were tested more than once. The median sampling interval was 4 days among patients who were tested more than once (Supplementary Data 1).

We validated the performance of commercial kits, including specificity, sensitivity, and reproducibility. None of nine healthy controls, five patients infected with hepatitis B virus, or five patients with syphilis tested positive for S-IgM, S-IgG, RBD-IgM, RBD-IgG, N-IgM, or N-IgG (Supplementary Fig. 1). Among COVID-19 patients, the test signal continuously decreased with increasing dilution of the samples (Supplementary Fig. 2a). The coefficients of variation of technical replicates were minor, indicating good reproducibility (Supplementary Fig. 2b).

The level of total IgM was relatively low in the first week and gradually increased until the 5th week, followed by a continuous decrease to the initial level. The level of total IgG was higher than that of IgM during the first week and continuously increased until the 5th week, maintained a similar level until the 7th week, and then gradually decreased from the 8th week, but was still considerably elevated at the end of the observation period (12th week) (Fig. 1a). Consistent with previous observations[18], IgG rose rapidly during the early infection phase. The dynamic changes of antibody levels in patients who had measurements at more than three time points and at least once within the first 2 weeks are shown in Fig. 1b. The profiles of antibody changes varied in different patients. For example, in Patient #880, the IgM level peaked on 15th day, and decreased between the 15th and 35th day, and then became negative 35 days after symptom onset. The IgG level of this patient peaked on the 20th day, was maintained until the 30th day, and then decreased but was still be positive on the 36th day. In contrast to Patient #880, the acute phase of infection in Patients #1096 and #1446 lasted <10 days after symptom onset, as the IgM level was negative and the IgG level peaked and started to decrease. We further quantified the total antibody levels of patients with confirmed SARS-CoV-2 infection and found that 39.6% of patients were IgM positive, and 70.8% were IgG positive within the first week after symptom onset (Fig. 1c). IgG could be detected in 95.3% of patients 5 weeks after symptom onset. Consistent with our observations, previous studies also found that the positive rate of IgG was higher than IgM, unlike the previous experiences from some other infectious diseases, including SARS-CoV-1[19]. Xu et al.[18] observed that the high level of IgG at the early stage of SARS-COV-2 infection was unique, compared with other viral infections which usually use IgM as an early marker for the acute phase. This phenomenon may result from that some COVID-19 patients are asymptomatic at the beginning of infection[20,21].

## Results

**Temporal profiles of total anti-SARS-CoV-2 antibodies**. To explore the temporal dynamic changes of immune response after

**The antibody response pattern in patients with different severity**. We also compared the levels of S-, RBD-, N-specific

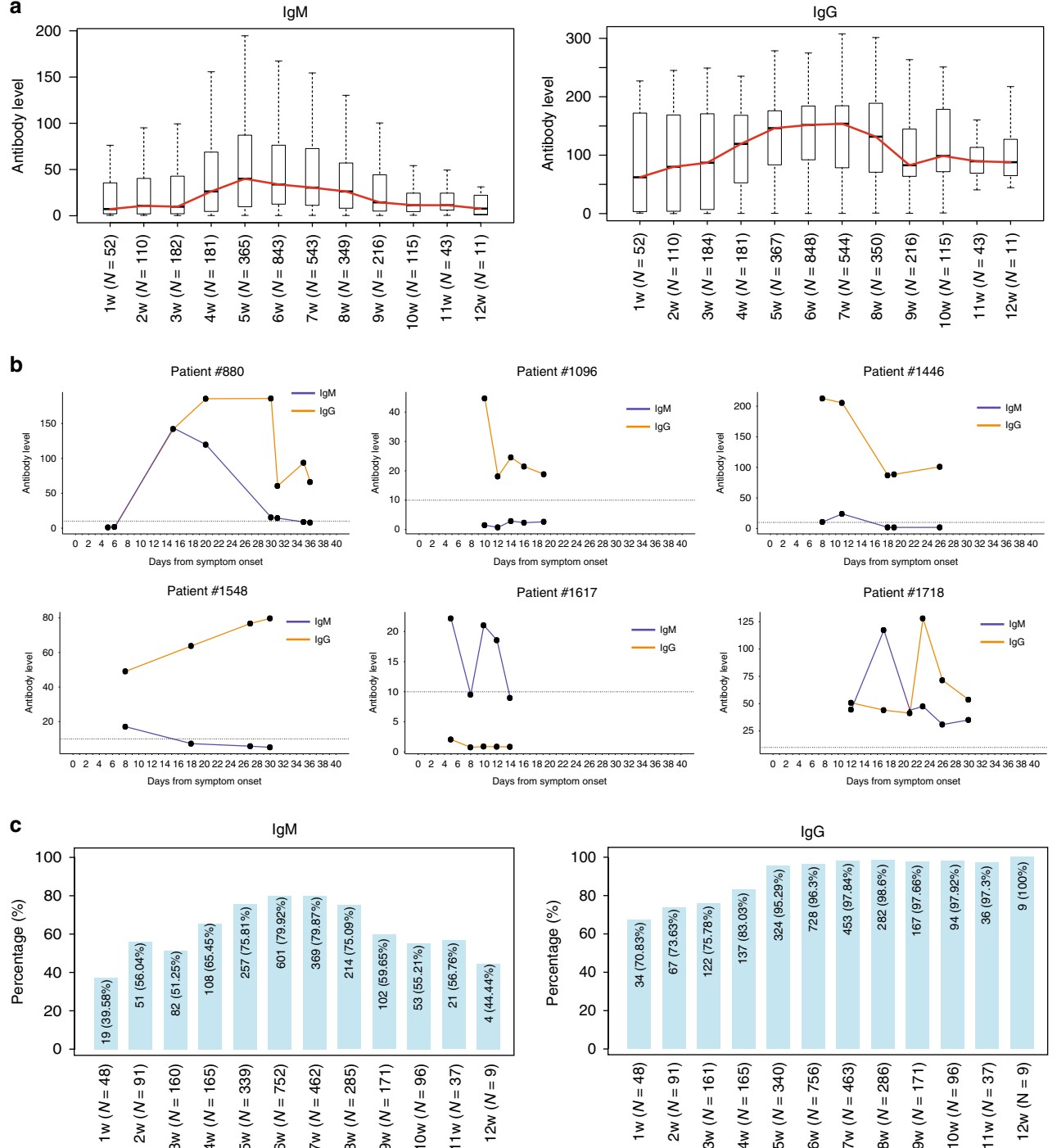

**Fig. 1 Temporal dynamic changes of the total anti-SARS-CoV-2 IgM and IgG. a** The total anti-SARS-CoV-2 IgM and IgG level of confirmed COVID-19 patients from the first to 12th week after symptom onset. Horizontal lines in the boxplots represent the median, the lower, and the upper hinges correspond to the first and third quartiles, and the whiskers extend from the hinge up to 1.5 times the interquartile range from the hinge. The red line based on the median is used to profile the variation tendency. **b** The temporal dynamic changes of the antibody levels in six representative patients. The *X* axis represents the days since symptom onset, and *Y* axis represents the antibody level. **c** The prevalence of total anti-SARS-CoV-2 IgM and IgG. The *X* axis indicates the weeks after symptom onset, and the *Y* axis shows the antibody-prevalence among confirmed COVID-19 patients.

antibodies between COVID-19 patients with mild/moderate and severe/critical disease. As shown in Supplementary Fig. 3, in COVID-19 patients with mild/moderate disease, the level of S-, RBD-, and N-specific IgM reached a high level in the second week after symptom onset, while it took 3 weeks for the severe/critical COVID-19 patients to reach comparable antibody levels. In COVID-19 patients with mild/moderate disease, S-, RBD-, and

N-specific IgG became detectable within the first 2 weeks after symptom onset, then sharply increased in the third week, and peaked in the fifth or sixth week. However, in COVID-19 patients with severe/critical disease, the level of S-, RBD-, and N-specific IgG only became detectable in the third week after symptom onset and peaked in the seventh week. The IgG levels remained at a high level until the end of the observation period (12th week).

The antibody response of patients with severe/critical COVID-19 lagged behind those of patients with mild/moderate disease by ~1 week. In the first 3 weeks after disease onset, the S-, RBD-, and N-specific IgM/IgG levels were higher in patients with mild/moderate COVID-19 than in those with severe/critical disease,

implying the weak antibody response in the early stage of infection may be associated with the disease progression. In the middle and late stages of infection (7–10 weeks after onset), the IgG levels were significantly higher in patients with severe/critical COVID-19 than in those with mild/moderate disease. The antibody levels were compared among COVID-19 patients who experienced mild/moderate and severe/critical disease, at the time of admission, hospitalization, and discharge. The total IgG, S-, RBD-, and N-specific IgG levels in patients with severe/critical COVID-19 patients were lower than those of patients the mild/moderate disease on admission, but these levels increased during hospitalization and on discharge (Table 1). The RBD-specific IgG level was approximately 1.5-fold higher in the patients with severe/critical disease than in those with mild/moderate disease during hospitalization ($P = 0.006$, two-sided Wilcoxon rank-sum test), and this ratio increased to 1.8 on discharge ($P = 0.001$, two-sided Wilcoxon rank-sum test). The S-specific IgG levels were also significantly higher in patients with severe/critical COVID-19 during hospitalization and on discharge ($P = 0.005$ and $P = 0.007$, respectively, two-sided Wilcoxon rank-sum test, Table 1).

Compared with patients with mild/moderate COVID-19, patients with severe/critical disease initially experienced a later antibody response but they developed a stronger response during the middle and late stages of infection. These results highlight the importance of timely medical intervention for patients with severe/critical diseases.

**Older COVID-19 patients have higher IgG levels during hospitalization.** We compared the antibody levels among patients of different age groups (<40 years, 40–65 years, and >65 years) (Table 2). On admission, the RBD-specific antibody level in patients aged >65 years was relatively lower than in younger and middle-aged patients, while the N-specific antibody level of older patients was higher. It has been reported that a high viral load in early infection may cause higher N-specific antibody levels because of the high immunogenic activity of N protein[22], but there was no evidence that N-specific antibodies can block viral replication. The S-, RBD-, and N-specific IgG levels were gradually elevated along with the age increase during hospitalization and on discharge ($P < 0.05$, two-sided Wilcoxon rank-sum test). For example, the median level of RBD-specific antibody in younger, middle-aged, and older patients was 7.7 AU/mL, 22.4 AU/mL, and 30.7 AU/mL, respectively (younger vs. middle-aged: $P < 0.001$, middle-aged vs. older: $P = 0.003$, two-sided Wilcoxon rank-sum test) during hospitalization.

**Table 1 SARS-COV-2-specific antibody levels in COVID-19 patients with different severity.**

|  | Mild/Moderate | Severe/critical | P value* |
|---|---|---|---|
| *On admission, median (IQR)* |  |  |  |
| Total SARS-CoV-2 IgM | 26.9 (8.5–63.8) | 23.6 (7.5–66.9) | 0.3 |
| Total SARS-CoV-2 IgG | 153.2 (88.3–184.8) | 148.8 (80.1–182.3) | 0.1 |
| RBD-specific IgM | 1.6 (0.8–3.7) | 1.5 (1.0–5.5) | 0.8 |
| RBD-specific IgG | 23.2 (9.5–47.3) | 18.5 (6.4–29.6) | 0.3 |
| S-specific IgM | 2.1 (1.5–4.4) | 2.6 (1.3–5.9) | 0.8 |
| S-specific IgG | 28.9 (11.4–41.7) | 17.2 (11.1–24.5) | 0.2 |
| N-specific IgM | 0.8 (0.3–1.6) | 1.1 (0.5–2.7) | 0.6 |
| N-specific IgG | 18.8 (7.1–26.8) | 15.7 (9.4–21.8) | 0.5 |
| *Highest level during hospitalization, median (IQR)* |  |  |  |
| Total SARS-CoV-2 IgM | 30.9 (9.7–66.9) | 37.5 (12.0–82.0) | 0.01 |
| Total SARS-CoV-2 IgG | 154.1 (89.6–186.8) | 163.2 (107.7–191.7) | 0.002 |
| RBD-specific IgM | 1.2 (0.6–3.0) | 1.6 (0.9–3.5) | 0.3 |
| RBD-specific IgG | 18.9 (8.3–39.3) | 27.7 (14.2–48.4) | 0.006 |
| S-specific IgM | 1.9 (0.9–3.3) | 1.7 (0.9–4.1) | 0.8 |
| S-specific IgG | 20.9 (11.5–35.6) | 27.5 (17.0–40.4) | 0.005 |
| N-specific IgM | 0.7 (0.3–1.5) | 0.7 (0.3–2.1) | 0.6 |
| N-specific IgG | 13.5 (6.9–27.2) | 17.1 (8.3–31.7) | 0.1 |
| *On discharge, median (IQR)* |  |  |  |
| Total SARS-CoV-2 IgM | 25.8 (6.6–60.9) | 21.6 (7.0–63.5) | 0.9 |
| Total SARS-CoV-2 IgG | 122.0 (69.0–174.9) | 114.2 (65.1–174.0) | 1 |
| RBD-specific IgM | 1.1 (0.5–2.3) | 1.3 (0.7–3.0) | 0.2 |
| RBD-specific IgG | 13.7 (6.0–29.9) | 24.4 (13.1–43.7) | 0.001 |
| S-specific IgM | 1.3 (0.7–2.7) | 1.3 (0.7–3.0) | 1 |
| S-specific IgG | 18.1 (7.6–27.9) | 23.8 (14.6–37.3) | 0.007 |
| N-specific IgM | 0.6 (0.3–1.5) | 0.6 (0.2–1.6) | 0.9 |
| N-specific IgG | 10.3 (3.6–29.4) | 15.3 (6.7–26.2) | 0.1 |

IgG immunoglobulin G, IgM immunoglobulin M, IQR interquartile range, N nucleoprotein, RBD receptor-binding domain, S spike, SARS-CoV-2 severe acute respiratory syndrome coronavirus 2.
*P values calculated using a two-sided Wilcoxon rank-sum test.

**Table 2 SARS-COV-2-specific IgG levels in COVID-19 patients according to age.**

|  | Age (<40 yrs) | Age (40–65 yrs) | Age (>65 yrs) | P value 1* | P value 2# |
|---|---|---|---|---|---|
| *On admission, median (IQR)* |  |  |  |  |  |
| RBD | 11.6 (3.6–20.0) | 20.4 (9.2–43.9) | 9.8 (5.5–39.9) | 0.2 | 0.5 |
| S | 21.3 (5.5–38.5) | 18.9 (11.3–35.4) | 20.7 (8.1–37.8) | 0.9 | 0.8 |
| N | 8.4 (5.4–14.1) | 15.7 (6.2–25.2) | 20.3 (10.4–28.0) | 0.4 | 0.3 |
| *Highest level during hospitalization, median (IQR)* |  |  |  |  |  |
| RBD | 7.7 (2.5–15.8) | 22.4 (13.4–37.5) | 30.7 (14.0–54.4) | <0.0001 | 0.003 |
| S | 8.9 (4.6–20.4) | 22.9 (12.5–32.3) | 28.0 (17.2–45.1) | <0.0001 | 0.0004 |
| N | 7.2 (4.1–19.7) | 12.1 (6.5–21.3) | 19.4 (9.3–-38.1) | 0.04 | <0.0001 |
| *On discharge, median (IQR)* |  |  |  |  |  |
| RBD | 7.1 (5.4–17.6) | 20.4 (11.1–37.1) | 27.4 (12.3–45.2) | 0.006 | 0.1 |
| S | 11.5 (5.9–21.1) | 23.7 (12.5–34.3) | 25.2 (15.8–40.8) | 0.01 | 0.1 |
| N | 8.2 (4.2–17.8) | 10.9 (4.7–27.2) | 16.2 (7.5–28.9) | 0.4 | 0.1 |

*P values of two-sided Wilcoxon rank-sum test between patients aged younger than 40 years and 40–65 years.
#P values of two-sided Wilcoxon rank-sum test between patients aged 40–65 years and older than 65 years.
COVID-19 coronavirus disease 2019, IgG immunoglobulin G, N nucleoprotein, RBD receptor-binding domain, S spike, SARS-CoV-2 severe acute respiratory syndrome coronavirus 2.

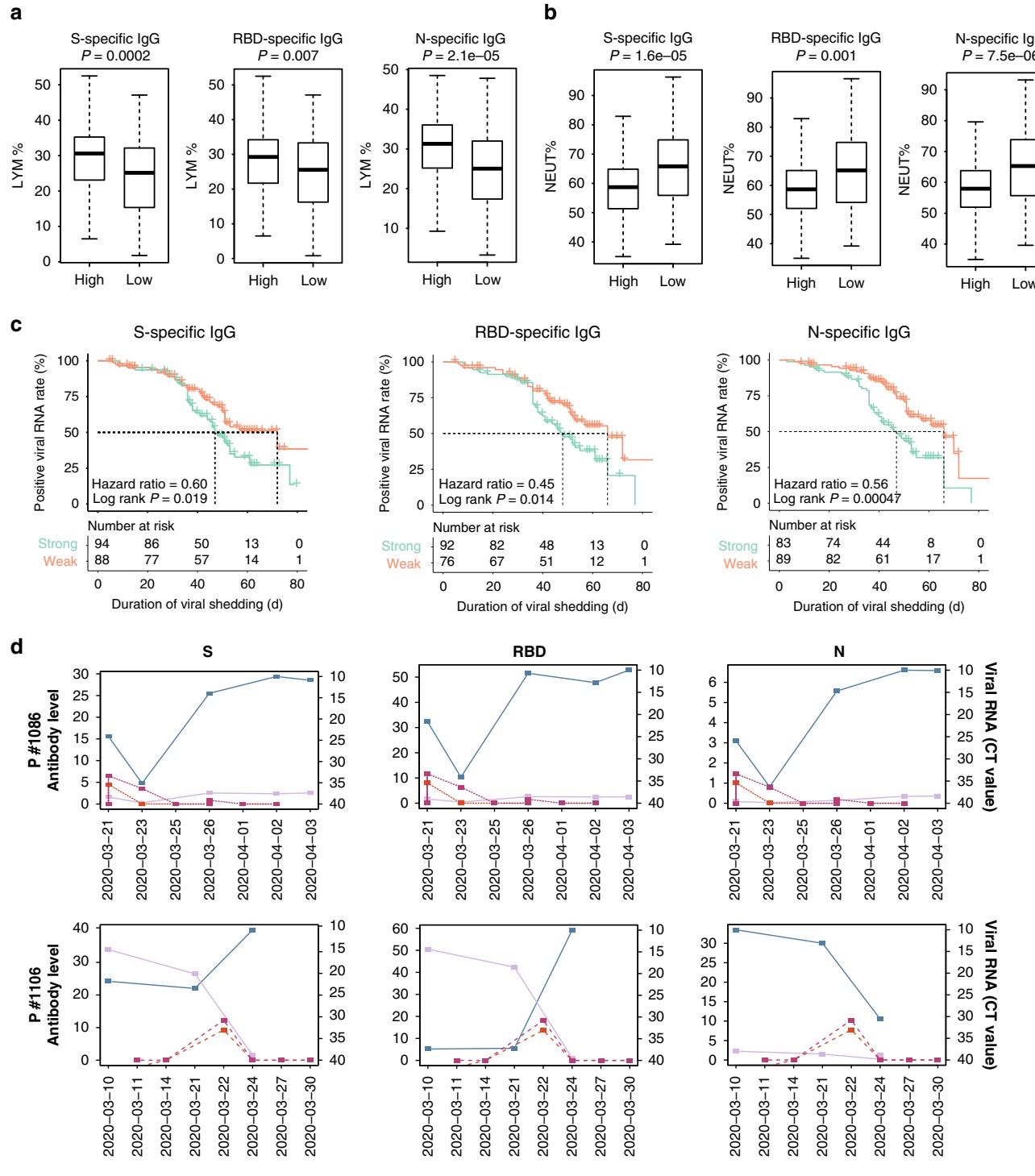

**Fig. 2 Relationship between the levels of IgG and the clinical outcome of COVID-19 patients.** The percentage of lymphocytes (**a**) and neutrophils (**b**) in severe/critical COVID-19 patients with different N-, RBD-, and S-specific IgG levels. Antibody levels and lymphocyte/neutrophil percentages were measured on the same day, with 112 sets of measurements for both the low- and high-antibody groups. Horizontal lines in the boxplots represent the median, the lower, and the upper hinges correspond to the first and third quartiles, and the whiskers extend from the hinge up to 1.5 times the interquartile range from the hinge. $P$ values were calculated with a two-sided Wilcoxon rank-sum test. **c** Kaplan–Meier analysis of the viral shedding time in patients with strong and weak antibody responses. The $X$ axis represents the duration of viral shedding (days). The $Y$ axis represents the positive rate of viral RNA. $P$ values were calculated with the log-rank test. **d** The dynamic changes in antibody levels and virus RNA load in Patients #1086 and #1106. The $X$ axis represents the detection date. The $Y$ axis on the left represents the antibody level, and the $Y$ axis on the right represents the cycle threshold (CT) value of PCR for the detection of viral RNA load. A CT value <40 was defined as SARS-CoV-2 viral positive. Blue dots represent IgG levels, purple dots represent IgM levels. The *ORF1ab* and *N* genes of SARS-CoV-2 were represented as pink and orange dots, respectively.

**Table 3 Comparison of the SARS-COV-2-specific antibody levels between SARS-CoV-2 RNA-positive and -negative patients.**

|  | Virus-positive | Virus-negative | *P* value* |
|---|---|---|---|
| RBD-specific IgM level | 1.3 (0.6–3.4) | 1.2 (0.7–3.3) | 1 |
| RBD-specific IgG level | 13.3 (6.6–23.5) | 28.2 (7.4–44.2) | 0.03 |
| S-specific IgM level | 1.6 (0.7–4.1) | 1.3 (0.6–2.7) | 0.4 |
| S-specific IgG level | 12.8 (9.7–30.9) | 25.1 (13.4–40.0) | 0.07 |
| N-specific IgM level | 0.5 (0.2–1.1) | 0.9 (0.4–2.2) | 0.02 |
| N-specific IgG level | 12.1 (5.1–25.4) | 12.4 (4.9–31.6) | 0.5 |

Only patients who tested both virus load and antibody level on the same day were included.
*IgG* immunoglobulin G, *IgM* immunoglobulin M, *N* nucleoprotein, *RBD* receptor-binding domain, *S* spike, *SARS-CoV-2* severe acute respiratory syndrome coronavirus 2.
*P values calculated using a two-sided Wilcoxon rank-sum test.

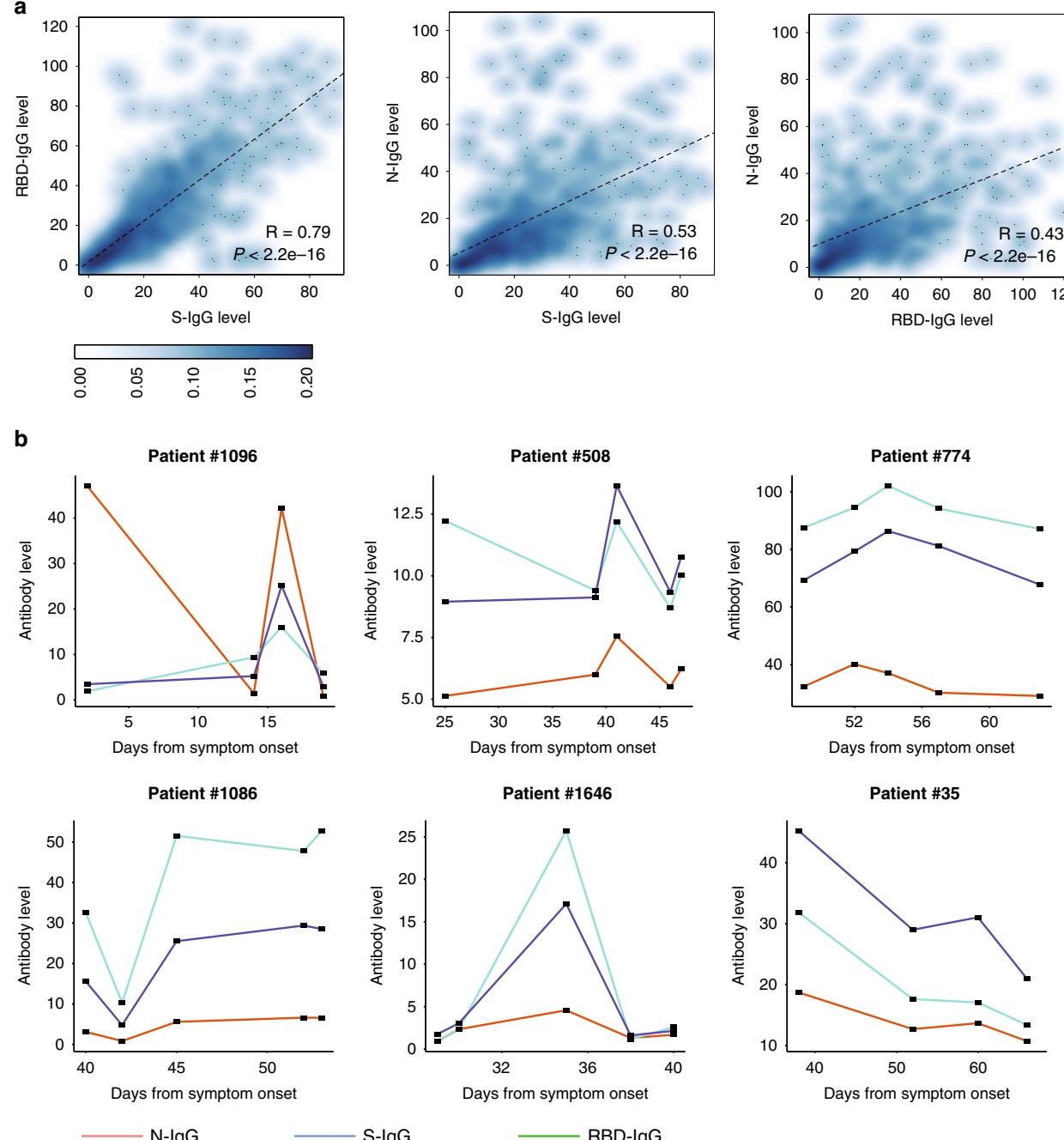

**Fig. 3 Correlations between S-, RBD-, and N-IgG levels. a** Scatter plots of the pair-wise correlations among S-, RBD-, and N-IgG levels. Each point represents the IgG level of one sample. *P* values were calculated with the Pearson correlation test. The density of points is shown by color. **b** Examples of the dynamic changes in the S-, RBD-, and N-IgG levels.

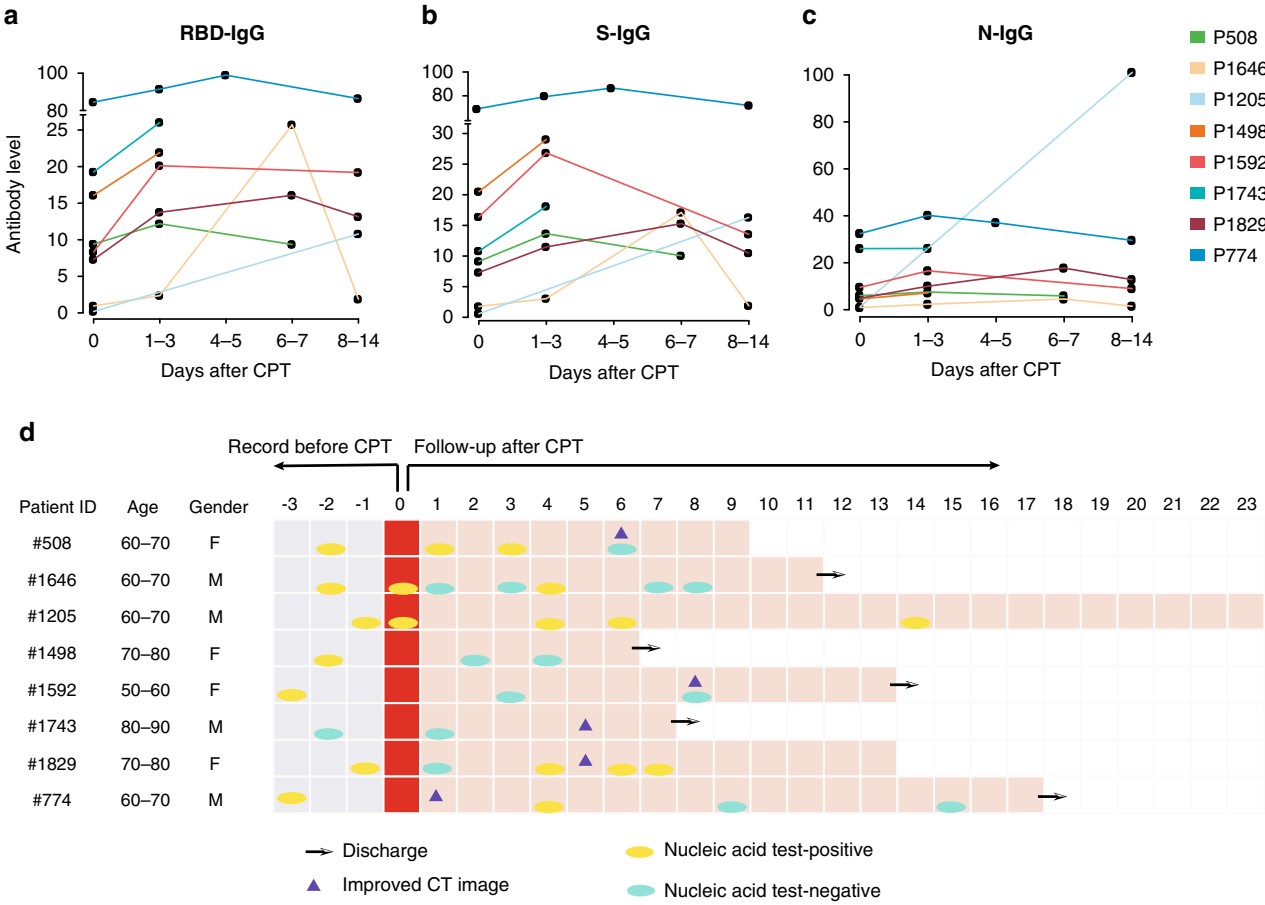

**Fig. 4 Dynamic changes in the IgG levels after convalescent plasma transfusion. a–c** The *X* axis represents the days since convalescent plasma transfusion therapy, and the *Y* axis represents the S-, RBD-, and N-specific IgG levels. Different colored lines represent the dynamic changes in antibody levels in different patients. **d** The clinical assessment of patients after receiving COVID-19 convalescent plasma transfusion therapy. Arrows represent the patients' discharge date. Triangles represent radiological improvements. Yellow and green dots represent the date in which the RT-PCR tests for SARS-CoV-2 RNA were positive and negative, respectively.

**IgG levels are associated with clinical outcome of COVID-19 patients**. As the antibody levels significantly increased in the severe/critical COVID-19 patients during hospitalization, we attempted to evaluate the function of these antibodies in the recovery of the severe/critical COVID-19 patients. The total IgM and IgG levels were tested in 46 serum samples from 21 non-survivors. We compared the average antibody levels between survivors and nonsurvivors by calculating the average IgG/IgM level of each patient. The antibody levels in nonsurvivors were significantly lower than those in survivors ($P = 0.01$ and $P = 0.06$ for IgM and IgG, respectively, two-sided Wilcoxon rank-sum test, Supplementary Fig. 4), suggesting that the antibody response played important roles in helping the severe/critical COVID-19 patients recover. It was not possible to compare the S-, RBD-, and N-specific antibody levels in nonsurvivors and survivors due to a lack of data in nonsurvivors.

According to the latest research, the ratio between neutrophil and lymphocyte percentage is an important prognostic indicator in patients with COVID-19[23]. Patients with a lower lymphocyte percentage and higher neutrophil percentage are more likely to have a poor outcome[23]. To evaluate whether the antibody levels are correlated with the neutrophil and lymphocyte percentage, we compared the results of antibody tests and routine blood tests performed on the same day in patients with severe/critical COVID-19 (Fig. 2a, b). The percentage of lymphocytes was significantly lower in patients with a low level of S-IgG, RBD-IgG, and N-IgG than in patients with higher IgG levels (S-specific IgG:

$P < 0.001$, RBD-specific IgG: $P = 0.007$, N-specific IgG: $P < 0.001$, two-sided Wilcoxon rank-sum test). Nevertheless, the neutrophil percentage was higher in patients with low IgG levels (S-specific IgG: $P < 0.001$, RBD-specific IgG: $P = 0.001$, N-specific IgG: $P < 0.001$, two-sided Wilcoxon rank-sum test). These results further revealed that the S-, RBD-, and N-specific IgG may prevent disease progression.

**IgG levels are associated with duration of viral shedding**. To determine the relationship between SARS-COV-2 clearance and antibody response, we analyzed the test results of patients who underwent viral RNA load and antibody testing on the same day and compared the S-, RBD-, and N-specific antibody levels in patients with detectable and undetectable SARS-CoV-2 RNA results. The S-specific and RBD-specific antibody levels were twofold higher in patients who were SARS-CoV-2 RNA negative than in those who were RNA positive (Table 3). The median of S-specific and RBD-specific IgG levels were 12.8 AU/mL vs. 25.1 AU/mL ($P = 0.07$, two-sided Wilcoxon rank-sum test), and 13.3 AU/mL vs. 28.3 AU/mL ($P = 0.03$, two-sided Wilcoxon rank-sum test) in RNA-negative and RNA-positive patients, respectively, indicating the important role of these antibodies in viral clearance.

We compared the time of viral shedding in patients with different levels of antibodies. The maximum level of S-, RBD-, and N-specific antibodies of each patient was calculated, and the top and bottom 33% of patients were categorized as strong-

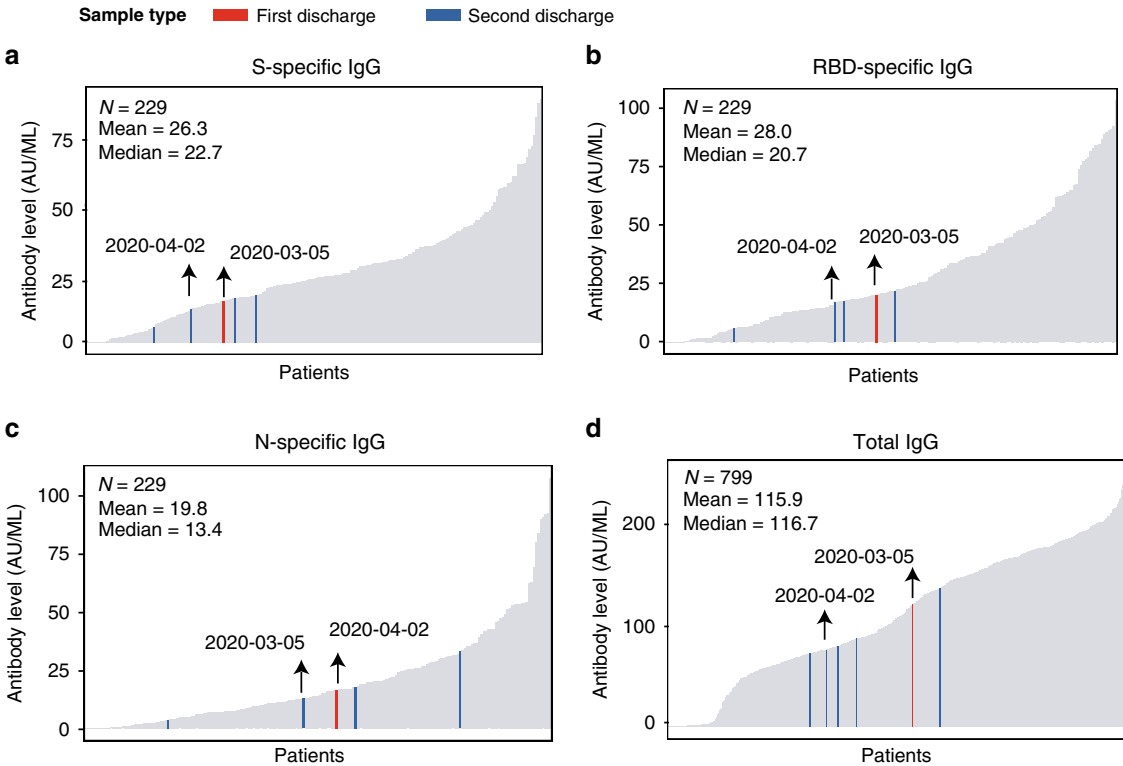

**Fig. 5 Level of IgG on discharge.** Barplot displays of the level of S-specific (**a**), RBD-specific (**b**), N-specific (**c**), and total IgG (**d**) in patients on discharge. The red bar indicates the time of the first discharge, and the blue bar indicates the time of the second discharge. The black arrows indicate the antibody levels of Patient #515.

responders and weak-responders. The duration of virus shedding was defined from the symptom onset to the date of the last positive nasopharyngeal swab. Patients with a weak antibody response had a significantly longer duration of viral shedding than those with a strong antibody response (log-rank $P = 0.019$, 0.014, and <0.001 for S-, RBD-, and N-specific IgG, respectively, Fig. 2c). This trend was observed in both mild/moderate and severe/critical patients (Supplementary Fig. 5). Although three kinds of antibodies all showed correlations with the duration of viral shedding, this result should be interpreted with caution. It does not indicate that these three antibodies all have the effect of neutralizing the virus, because the generation of these antibodies may be accompanying. We analyzed the correlation between the levels of S-IgG, RBD-IgG, and N-IgG (Fig. 3a). The S-IgG and RBD-IgG levels were highly positive-related because the RBD is part of the S1 unit of the S protein ($R = 0.79$), while the S-IgG/RBD-IgG and N-IgG levels were less strongly correlated ($R = 0.53$ between S-IgG and N-IgG, and $R = 0.43$ between RBD-IgG and N-IgG). Furthermore, the levels of these three IgG changed jointly in most patients, as illustrated by Patients #508 and #1646 (Fig. 3b). By observing the dynamic changes of viral RNA load and antibody levels in patients with consecutive results, we found that there was an association between viral shedding and antibody levels (Fig. 2d and Supplementary Fig. 6). For example, in Patient #1086, viral shedding was accompanied by an increase of antibody levels, while Patient #1106 became SARS-CoV-2 RNA negative after the level of S-IgG and RBD-IgG increased, despite a lack of increase in N-IgG.

**Antibody dynamic changes after convalescent plasma transfusion.** Because the plasma of patients who have recovered from infection may contain neutralizing antibodies against the virus, human convalescent plasma therapy (CPT) is an option for the prevention and treatment of COVID-19[24,25]. We analyzed the changes of levels of S-, RBD-, and N-specific IgG in 1–3 days, 4–5 days, 6–7 days, and 8–14 days after CPT in eight patients who received consecutive antibody detection both before and after CPT (Fig. 4). The level of RBD-specific and S-specific IgG increased in seven of the eight patients within 3 days after CPT, but the level of N-specific IgG had no obvious change. Five out of seven patients who were SARS-CoV-2 RNA positive before CPT were RNA negative or showed signs of radiologic improvement within 3 days after CPT, indicating the potential role of CPT in providing immediate immunity for recipients. Some retrospective studies or clinical trials reported that CPT therapy could provide an improvement in the symptoms and a reduction in mortality among patients with severe/critical COVID-19 disease[15,25,26]. The increase of anti-SARS-CoV-2 antibody levels is one of the most important reasons for the effectiveness of CPT therapy.

**Low antibody levels on discharge increase the risk of re-detectable viral RNA.** Patients who show obvious improvement in respiratory symptoms and two consecutive negative RNA tests on nasopharyngeal swabs can be discharged. However, patients who meet these discharge criteria may not have sufficient antibodies to prevent them from re-infection. We observed that a considerable number of patients had low antibody levels at the time of discharge. Specifically, among the patients who were tested for antibodies within 3 days before discharge, 11.65% (74/635) were negative for total IgG, and 5.70% (11/193), 7.77% (15/193), and 9.33% (18/193) were negative for S-IgG, RBD-IgG, and N-IgG, respectively. Given the important role of S-IgG and RBD-IgG in neutralizing SARS-CoV-2, it is reasonable to speculate that the patients who discharged without these protective antibodies may have a high risk of re-infection. As shown in Fig. 5, Patient #515 was discharged on March 5, 2020, with a relatively low level

of S-, RBD-, and N-specific antibodies. This patient had a re-detectable positive RNA test result and was readmitted to the hospital for treatment on March 15. He was discharged for the second time on April 2, still with a low level of antibodies. Another three patients who also showed re-detectable positive RNA test after discharge did not have their antibody levels tested prior to their first discharge and they still had a low level of antibodies at the time of their second discharge. Therefore, patients with undetectable or low antibody levels may need close monitoring after discharge.

## Discussion

By analyzing the laboratory data of patients from Wuhan Huoshenshan Hospital, which is one of the biggest designated hospitals for COVID-19 in Wuhan, we profiled the temporal dynamic changes of the antibody levels in the 12 weeks following the disease onset, including total antibody and the S-, RBD-, N-specific antibodies. We found that compared with patients with mild/moderate disease, those with severe/critical disease experienced a delay in S-, RBD-, and N-specific IgG development of ~1 week. The IgG levels were significantly higher in older patients and those with more severe disease, indicating that these patients have greater activation of their immune defense during recovery. In addition, we found that the S- and RBD-specific IgG level was much higher in recovered COVID-19 patients who were SARS-CoV-2 negative, indicating that antibodies play an important role in viral clearance. The lower S-, RBD- and N-IgG levels were associated with lower lymphocyte percentage, higher neutrophil percentage, and longer duration of viral shedding. Importantly, the patients who have low levels of antibodies at the time of hospital discharge may have a high risk of developing re-detectable SARS-CoV-2 RNA on RT-PCR testing after recovery, demonstrating the prognostic value of antibody level for discharged COVID-19 patients. Overall, our results suggested that these IgG, especially S-specific or RBD-specific IgG played an important role in viral clearance and recovery of COVID-19 patients.

It is widely recognized that IgM provides the first line of defense during viral infections, while the production of IgG lags behind IgM, and provides long-term immunity and memory[7]. According to a previous report about SARS in 2003, IgM could be detected in patients' blood 3–6 days after disease onset, while IgG could be detected 8 days after disease onset[27]. Our observations showed that the total anti-SARS-Cov-2 IgG level was already at a relatively high level in the first week after disease onset, consistent with a previous study that found an early and high level of IgG response against SARS-CoV-2[18]. The high positive rate of IgG at the early stage of SARS-CoV-2 infection may because some COVID-19 patients are asymptomatic during the early stage of infection[20,28,29]. According to a recent report, 97.5% of people who develop symptoms do so within 11.5 days[21]. Clinicians recorded the first day of the patients' symptoms, such as fatigue, fever, cough, or diarrhea as the date of disease onset. However, the recorded onset date may be later than the date of infection due to asymptomatic infection, explaining why the IgG level was high during the first week after record onset.

Moreover, there have been reports that some patients have re-detectable positive RNA tests after recovery[30], raising questions about the patients' condition on discharge. According to the latest version of "Diagnosis and Treatment Protocol for Novel Coronavirus Pneumonia of China", patients who meet the following criteria can be discharged: (1) a return of body temperature to normal for more than three days; (2) an obvious improvement in respiratory symptoms; (3) pulmonary imaging shows obvious absorption of inflammation; and (4) two consecutive negative nucleic acid tests on respiratory tract samples, such as sputum and nasopharyngeal swabs (measured on samples collected at least 24 h apart). However, our study shows that patients with re-detectable positive RNA test results tended to have low antibody levels on discharge, highlighting the importance of antibody testing prior to discharge. In our opinion, there are two possible reasons for re-detectable positive RNA test results among recovered COVID-19 patients. First, studies have shown that SARS-CoV-2 mainly infects the lower respiratory tract[31], but the collection of the bronchoalveolar lavage requires skilled operators and specific devices, exposing the clinicians to a high risk of infection. So, the nasopharyngeal swab samples are used to assess the viral load, the sample quality may lead to false-negative results. Second, the immune defense system of some discharged patients was relatively weak. Lacking sufficient antibody may make patients susceptible to re-infection. Thus, patients with negative nasopharyngeal swabs but low S- or RBD-specific IgG levels need to raise discharge criterion or strengthen protection and monitoring after discharge.

There are some limitations in this study. First, just as the cumulative observations showed, the false-negative of viral RNA test of nasopharyngeal swabs may be inevitably included in this study. Second, as one of the largest designated hospitals which were built specifically for COVID-19 pandemic in Wuhan, Huoshenshan Hospital admitted some patients transferred from other non-designated hospitals when it opened in early February; hence, it is difficult to reconstruct the temporal development of antibodies in these patients. Whether the antibodies against S, RBD, and N proteins develop at the same time or sequentially still needs further prospective studies. Third, ideally, this study should be based on continuous time-point detection, but it was not possible to conduct continuous systematic sample collection during the pandemic, and not all patients had continuous observation data, limiting the power of consecutively observing the antibody response in certain individuals.

In conclusion, our results suggest that antibodies against SARS-CoV-2 play an essential role in COVID-19 recovery. Tracking the dynamic changes of these antibodies can provide an important reference for diagnosis, monitoring, and prognosis of COVID-19. This study will shed new light on the development of novel agents, such as vaccine and monoclonal antibody for COVID-19.

## Methods

**Data collection**. We analyzed the laboratory test results of COVID-19 patients admitted from February 4 to March 30 to Wuhan Huoshenshan Hospital in China, in which patients were diagnosed based on the Diagnosis and Treatment Protocol for Novel Coronavirus Pneumonia released by the National Health Commission (Version 6 or trail version 7, https://www.who.int/docs/default-source/wpro—documents/countries/china/covid-19-briefing-nhc/1-clinical-protocols-for-the-diagnosis-and-treatment-of-covid-19-v7.pdf). In this study, all enrolled patients were confirmed to be infected with SARS-CoV-2 using RT-PCR assays performed on nasopharyngeal swab samples (Supplementary Table 2). A total of 1850 patients who had been detected antibody levels and a clear record of symptom onset history were included (Supplementary Data 2). This cohort included 795 patients with mild or moderate disease and 1055 patients with severe or critical diseases. Data on the clinical characteristics and laboratory findings of all patients were extracted from the hospital electronic medical records (Supplementary Table 1 and Supplementary Data 3). This study was approved by the Medical Ethical Committee of Wuhan Huoshenshan Hospital (HSSLL011) and the Ethical Committee of Nanjing Medical University (2020-511). Written informed consent was obtained from each patient.

**Serum anti-SARS-CoV-2 antibodies assay**. Total SARS-CoV-2 serum IgM or IgG was measured at different time points using a magnetic chemiluminescence enzyme immunoassay (MCLIA) using commercially available kits (Shenzhen YHLO Biotech Co., Ltd., Shenzhen, China, catalog number 20200206). The magnetic beads of the MCLIA assay were coated with SARS-CoV-2 antigens containing N and S protein. All serum antibody tests were performed with an iFlash3000 automated MCLIA analyzer from Shenzhen YHLO Biotech Co., Ltd. Furthermore, 416 patients were tested for S-, RBD-, and N-specific IgM and IgG levels at

different time points by MCLIA using commercially available kits (Nanjing RealMind Biotech Co., Ltd., S-IgM catalog no.: R90120022001; S-IgG catalog no.: R90220022001; N-IgM catalog no.: R90520022001; N-IgG catalog no.: R90620022001; RBD-IgM catalog no.: R90320022001; RBD-IgG catalog no.: 90420022001). The antibody levels of all samples were measured following the manufacturer's instructions. All magnetic beads coated with antigens (recombinant SARS-CoV-2 N or S protein, or RBD) were dissolved in 100 mmol/L phosphate buffer (pH = 6.96 ± 0.10), at a concentration of 0.625 mg/mL. The combination of magnetic beads and antigen proteins was a two-step coupling method. First, the carboxyl groups on the surface of the beads were activated by 1-ethyl-3-[3-dime-thylaminopropyl] carbodiimide hydrochloride (EDC). Second, the EDC activation enabled the antigen proteins to attach to the magnetic beads. For each reaction, 10 μL of 20-fold diluted serum was added into the reaction cup, and then 20 μL of magnetic beads were added and incubated for 10 min at 37 °C. The antigen–antibody complex captured using the bead slurry was gently precipitated by a magnetic separation rack. The beads were then incubated with acridinium ester-labeled mouse anti-human IgM or IgG antibody and reacted with hydrogen peroxide in an excitation buffer. Chemiluminescence intensity was recorded in an ACL2800 chemiluminescence system (Nanjing RealMind Biotech Co., Ltd., Nanjing, China). Different dilutions of inactivated serum from the same COVID-19 patient were used as calibrators and a stable linear correlation between light intensity and relative antibody level was established by measuring its light intensity at different concentrations. Relative antibody levels were presented as the measured chemiluminescence values divided by the constant derived from the linear corre-lation (S/CO). S/CO > 1 was defined as positive and S/CO ≤ 1 as negative. All tests were performed under strict biosafety conditions, with the same batch of kit from the same manufacturer to ensure stable and uniform experimental conditions.

**RT-PCR detection for SARS-CoV-2.** The total nucleic acids were extracted from the nasopharyngeal swabs of patients. The ORF1ab and nucleocapsid (N) genes were detected by performing real-time PCR assay. The primer sequences are listed in Supplementary Table 2. The number of cycles (CT value) is used to measure the viral load. A higher CT value indicates to a lower viral load. A CT value <40 was defined as SARS-CoV-2 viral positive.

**Statistical analysis.** Continuous variables are reported as medians and inter-quartile ranges (IQR) and categorical variables are reported as frequencies and percentages. The Wilcoxon rank-sum test was used to compare groups. The Kaplan–Meier method is used to analyze the duration of viral shedding in patients with different antibody levels. All analyses are performed using R software (version 3.6.2).

**Reporting summary.** Further information on research design is available in the Nature Research Reporting Summary linked to this article.

## Data availability
All the data supporting the findings of this study are available within the article and its Supplementary Information files. Source data are provided with this paper.

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

## Acknowledgements

This study was supported by the National Natural Science Foundation of China (Grant Nos. 81572893, 81972358, and 91959113), Key Foundation of Wuhan Huoshenshan Hospital (Grant No. 2020[18]), Key Research & Development Program of Jiangsu Province (Grant Nos. BE2017733 and BE2018713), Medical Innovation Project of Logistics Service (Grant No. 18JS005), Basic Research Program of Jiangsu Province (Grant No. BK20180036). We thank Jinli Li for her advice for the validation of the assay.

## Author contributions

X.X., S.W., and Q.W. had full access to all of the data in the study and takes responsibility for the integrity of the data and the accuracy of the data analysis. K.L., B.H., M.W., and A.Z. contributed equally. Concept and design: X.X., S.W., and Q.W. Data collection: A.Z., Z.G., Zhihua Wang, Q.Q., and J.W. Data analysis and interpretation: K.L., M.W., B.H., L.L., Y.C., L.W., M.Z., J.L., Ziyu Wang, W.W., and W.L. Drafting of the paper: K.L., M.W., and B.B.

## Competing interests
The authors declare no competing interests.
