## [Peer Review File · Nature Communications]

Reviewer #1 (Remarks to the Author):

In the study by Li et al, the relative levels of IgM and IgG binding titers are measured in a cohort of 1850 COVID-19 patients from Wuhan China. The levels are compared among age groups, longitudinally and relative to symptom severity. The levels are also measured in survivors vs non-survivors, following convalescent plasma transfer. Interestingly the levels are measured in persons who were thought to be re-infected and they were found to have relatively low levels of antibody compared to the rest of the cohort. Throughout there is an assumption that the antibodies they are measuring are protective. Whether this is true is currently unknown and it is not demonstrated by this study.

Specific comments:

What is the dynamic range of the assay? Are there maximum and minimum limits of quantitation or detection?

No bead number or amount of slurry is reported so it's hard to tell whether the assays were all run in an identical manner.

Scant information is provided on the kits used, catalog numbers, how were the antigens produced, how were they coupled etc.

Error bars on Fig 1A look massive it's hard to tell if these trends are real.

There is an assumption made throughout that anti SARS-CoV-2 antibodies are protective and that among those the RBD-specific antibodies are the most protective. I presume that this is because the authors assume these will be neutralizing. It's not clear that antibodies have a protective effect in COVID-19 infection, nor is it clear whether antibodies targeting regions of the spike (eg RBD) are protective or even neutralizing. No neutralizing antibody data is provided to support this.

Several of the conclusions are based on this assumption. A few are listed here:

"Therefore, our results indicated that a fair number of virus-free patients were discharged with a low level of protective antibody"

"the median of RBD-specific IgG level was 13.3 AU/ML vs. 28.3 AU/ML between virus-positive and virus-negative status (P=0.03) indicating the important role of protective antibodies in virus clearance".

In Figure 3 serum antibody levels are measured after convalescent plasma transfer (CPT).

However it is impossible to determine whether the CPT had any effect on the measured titers, or on the outcome of their infection, yet the authors conclude that "these observations demonstrate that CPT therapy may provide the long-term antibody against the SARS-Cov-2 virus for patients and help the COVID-19 patients recover".

In the aged population the antibodies against the N protein are higher than the anti-N antibodies in the other age groups but the RBD specific levels are lower. The authors attribute this to "a weak immune defense response." The data don't support this statement this age cohort is clearly able to mount a robust antibody response, but it's just focused on the N protein.

Reviewer #2 (Remarks to the Author):

In this study, Li et al reports a comprehensive study on the kinetics of total antibody, spike protein (S)-, receptor-binding domain (RBD)-, and nucleoprotein (N)- specific IgM and IgG levels during SARS-CoV-2 infection based on samples from 1,850 patients. They found that the S-, RBD-, and N- specific IgG generation of severe/critical COVID-19 patients is one week later than mild/moderate cases, while the levels of these antibodies are 1.5-fold higher in severe/critical patients during hospitalization (P<0.01). The decrease of these IgG levels indicates the poor

outcome of severe/critical patients. Notably, they found that the patients who got re-infected had a low level of protective antibody on discharge. Together with previous studies, these findings inform the diagnosis, treatment, as well as vaccine development for COVID-19. However, the concerns below should be addressed by the authors.

1. Please introduce the criteria of patient recruitment. Additionally, how the patients were selected for this study as many patients were accepted for hospitalization in this hospital.
2. How many samples were taken from each patient? How the consecutive samples were collected? How about the interval of sampling?
3. The most interesting data related to the patient who was re-infected after discharge. Because the duration between the two hospitalizations is very close, how to exclude the persistent infection? As some patients were found to shed virus in a long term.
4. The authors should discuss why the antibody titer was lower in the early weeks in severe cases and obviously increased then? But the titers decreased sharply in the non-survivors?
5. The relationship between viral shedding and antibody can be discussed.
6. The sample size should be marked in legends of figure 2 and 3.
7. The demographic, clinical and laboratory data of the patients recruited should be disclosed by a table.
8. The limitation of the study should be presented in the discussion section.
9. The median time of antibody occurrence should be compared with other studies.

Reviewer #1

In the study by Li et al, the relative levels of IgM and IgG binding titers are measured in a cohort of 1850 COVID-19 patients from Wuhan China. The levels are compared among age groups, longitudinally and relative to symptom severity. The levels are also measured in survivors vs non-survivors, following convalescent plasma transfer. Interestingly the levels are measured in persons who were thought to be re-infected and they were found to have relatively low levels of antibody compared to the rest of the cohort. Throughout there is an assumption that the antibodies they are measuring are protective. Whether this is true is currently unknown and it is not demonstrated by this study.

Specific comments:

1. What is the dynamic range of the assay? Are there maximum and minimum limits of quantitation or detection?

Response: Our study tested the antibody levels (against spike, RBD, and nucleocapsid protein) in serum through chemiluminescence using different dilutions of inactivated serum from the same COVID-19 patient as calibrators and calculated the relative light unit (RLU) value to indicate the antibody levels. Given that chemiluminescence can semi-quantitatively measure the antibody levels of multiple samples rapidly and simultaneously, this measurement is widely used (Long et al., Nature Medicine, 2020; Pollán M et al., Lancet, 2020; Lerner AM et al. Immunity, 2020). Relative antibody levels were presented as the measured chemiluminescence values divided by the constant derived from the linear correlation (S/CO). There were no maximum and minimum limits for this detection. The summary of each IgG and IgM level in our dataset is provided below:

Response Table 1. Summary of antibody levels

	S-IgM	S-IgG	RBD-IgM	RBD-IgG	N-IgM	N-IgG
Min	0.0056	0.017	0.0031	0.0084	0.0026	0.0055
1st quarter	0.72	12.24	0.62	9.90	0.28	6.58
Median	1.52	22.72	1.26	21.49	0.58	13.46
3rd quarter	3.17	35.67	2.88	41.93	1.38	26.45
Max	33.94	88.82	50.52	119.84	33.38	103.67

2. No bead number or amount of slurry is reported so it's hard to tell whether the assays were all run in an identical manner.

Response: The antibody levels of all samples were measured using commercial kits following the manufacturer's instructions (Shenzhen YHLO Biotech Co., Ltd., catalog numbers 20200206; Nanjing RealMind Biotech Co., Ltd., S-IgM: R90120022001, S-IgG: R90220022001, N-IgM: R90520022001, N-IgG: R90620022001, RBD-IgM: R90320022001, RBD-IgG: 90420022001). According the instructions, all magnetic beads coated with antigens (SARS-CoV-2 N, S protein, or RBD) should be dissolved in 100 mmol/L phosphate buffer (pH=6.96 ± 0.10), and the concentration would be 0.625 mg/mL. For each reaction, 10 µL of 20-fold diluted serum was added to the reaction mixture, and then 20 µL of magnetic beads were added for mixed incubation for 10 min at 37°C. All serum samples were tested using the same batch of kits from the same manufacturer to ensure stable and uniform experimental conditions.

3. Scant information is provided on the kits used, catalog numbers, how were the antigens produced, how were they coupled etc.

Response: Thank you for your comments. According to the reviewers' suggestion, we revised the corresponding description of the detection of serum anti-SARS-CoV-2 antibodies as (**Lines 387-421**):

“Total SARS-CoV-2 serum IgM or IgG was measured at different time points using a magnetic chemiluminescence enzyme immunoassay (MCLIA) using commercially available kits (Shenzhen YHLO Biotech Co., Ltd., Shenzhen, China, catalog number 20200206). The magnetic beads of the MCLIA assay were coated with SARS-CoV-2 antigens containing N and S protein. All serum antibody tests were performed with an iFlash3000 automated MCLIA analyzer from Shenzhen YHLO Biotech Co., Ltd. Furthermore, 416 patients were tested for S-, RBD-, and N-specific IgM and IgG levels at different time points by MCLIA using commercially available kits (Nanjing RealMind Biotech Co., Ltd., S-IgM catalog no.: R90120022001; S-IgG catalog no.: R90220022001; N-IgM catalog no.: R90520022001;

N-IgG catalog no.: R90620022001; RBD-IgM catalog no.: R90320022001; RBD-IgG catalog no.: 90420022001). The antibody levels of all samples were measured following the manufacturer's instructions. All magnetic beads coated with antigens (recombinant SARS-CoV-2 N or S protein, or RBD) were dissolved in 100 mmol/L phosphate buffer (pH=6.96 ± 0.10), at a concentration of 0.625 mg/mL. The combination of magnetic beads and antigen proteins was a two-step coupling method. First, the carboxyl groups on the surface of the beads were activated by 1-ethyl-3-[3-dimethylaminopropyl] carbodiimide hydrochloride (EDC). Second, the EDC activation enabled the antigen proteins to attach to the magnetic beads. For each reaction, 10 µL of 20-fold diluted serum was added into the reaction cup, and then 20 µL of magnetic beads were added and incubated for 10 minutes at 37°C. The antigen-antibody complex captured using the bead slurry was gently precipitated by a magnetic separation rack. The beads were then incubated with acridinium ester-labeled mouse anti-human IgM or IgG antibody and reacted with hydrogen peroxide in an excitation buffer. Chemiluminescence intensity was recorded in an ACL2800 chemiluminescence system (Nanjing RealMind Biotech Co., Ltd., Nanjing, China). Different dilutions of inactivated serum from the same COVID-19 patient were used as calibrators and a stable linear correlation between light intensity and relative antibody level was established by measuring its light intensity at different concentrations. Relative antibody levels were presented as the measured chemiluminescence values divided by the constant derived from the linear correlation (S/CO). S/CO >1 was defined as positive and S/CO ≤1 as negative. All tests were performed under strict biosafety conditions, with the same batch of kit from the same manufacturer to ensure stable and uniform experimental conditions.”

4. Error bars on Fig 1A look massive it's hard to tell if these trends are real.

Response: Figure 1A describes the weekly antibody levels of all enrolled patients. Unlike the replicate experiments in the laboratory, the antibody levels varied highly among different patients. Published data on the antibody response against SARS-CoV-2 also indicates high heterogeneity among patients (Kelvin Kai-Wang To et al., Lancet Infect Dis, 2020, 20: 565–74; Long et al., Nature Medicine, 2020: 1-4; Juno et al. Nature Medicine, 2020: 1-7; Xu et al.

Nature Medicine, 2020: 1-3). Despite the large differences in antibody levels among patients, our study provides a comprehensive analysis of the dynamic trends of IgM and IgG by assessing 3063 serum samples from 1850 patients. The general trends of total IgM and IgG levels were profiled based on the median level in each week (**Response Figure 1A**). Furthermore, the dynamic changes in antibody levels in patients with more than 3 total detections, at least one of which was within the first two weeks, are also shown in **Response Figure 1B**. Our results showed that the profiles of antibody changes varied in different patients. For example, in Patient #880, the IgM level peaked on 15th day, and decreased between the 15th and 35th day, and then became negative 35 days after symptom onset. The IgG level of this patient peaked on the 20th day, was maintained until the 30th day, and then decreased but was still be positive on the 36th day. In contrast to Patient #880, the acute phase of infection in Patients #1096 and #1446 lasted less than 10 days after symptom onset, as the IgM level was negative and the IgG level peaked and started to decrease. We added the following to the revised manuscript (**Lines 123-138**):

“The level of total IgM was relatively low in the first week, and gradually increased until the fifth week, followed by a continuous decrease to the initial level. The level of total IgG was higher than that of IgM during the first week, and continuously increased until the 5th week, maintained a similar level until the 7th week, and then gradually decreased from the 8th week, but was still considerably elevated at the end of the observation period (12th week) (**Figure 1A**). Consistent with previous observations¹⁸, IgG rose rapidly during the early infection phase. The dynamic changes of antibody levels in patients who had measurements at more than three time points and at least once within first two weeks are shown in **Figure 1B**. The profiles of antibody changes varied in different patients. For example, in Patient #880, the IgM level peaked on 15th day, and decreased between the 15th and 35th day, and then became negative 35 days after symptom onset. The IgG level of this patient peaked on the 20th day, was maintained until the 30th day, and then decreased but was still be positive on the 36th day. In contrast to Patient #880, the acute phase of infection in Patients #1096 and #1446 lasted less than 10 days after symptom onset, as the IgM level was negative and the IgG level peaked and started to decrease.”

Response Figure 1 (Related to Figure 1 in revised manuscript). Temporal dynamic changes in the total IgM and IgG levels in confirmed COVID-19 cases. (A) The total IgM and IgG level in COVID-19 patients from the 1st to the 12th week after symptom onset. Each boxplot depicts the level of antibodies, and whiskers represent the maximum and minimum. The red line based on the median value represents the variation tendency. (B) The temporal dynamic changes in the antibody levels in six patients. The x-axis represents the days after symptom onset, and the y-axis represents the antibody level.

5. There is an assumption made throughout that anti SARS-CoV-2 antibodies are protective and that among those the RBD-specific antibodies are the most protective. I presume that this is because the authors assume these will be neutralizing. It's not clear that antibodies have a protective effect in COVID-19 infection, nor is it clear whether antibodies targeting regions of the spike (e.g. RBD) are protective or even neutralizing. No neutralizing antibody data is provided to support this. Several of the conclusions are based on this assumption. A few are listed here: "Therefore, our results indicated that a fair number of virus-free patients were discharged with a low level of protective antibody"; "the median of RBD-specific IgG level was 13.3 AU/ML vs. 28.3 AU/ML between virus-positive and virus-negative status (P=0.03) indicating the important role of protective antibodies in virus clearance".

Response: Thank you for pointing this out. Cumulative studies suggest that antibodies targeting S protein, and especially the RBD domain, are the primary neutralizing antibodies. The RBD domain in the S1 subunit can interact with human cells expressing angiotensin-converting enzyme 2 (ACE2) and induce virus entry (Jiang et al., Trends in Immunology, 2020). The neutralizing antibodies often target the RBD domain of the S protein to block the interaction between the virus and the host receptor (Cao, Nat Rev Immunol, 2020). Antibodies against S protein, and especially the RBD domain of SARS-CoV-2, serve as a target for vaccine and therapeutic development (Du et al, Nature Reviews, 2009). Wu et al. developed a full library of human single-domain antibody to screen for SARS-CoV-2-specific single-domain antibodies. The neutralizing effect of these antibodies was measured using a pseudovirus neutralizing assay, and antibodies targeting five different epitopes on RBD were recognized to have the ability to neutralize SARS-CoV-2 (Wu et al., Cell Host & Microbe, 2020). Notably, some recent studies have reported significant progress in the development of COVID-19 therapy, based on the S protein and RBD domain. Chi et al. isolated and characterized a neutralizing monoclonal antibody binding to the S protein of SARS-CoV-2 from ten convalescent COVID-19 patients (Science, 22 Jun 2020, DOI: 10.1126/science.abc6952). Some newly developed vaccines have also been designed based on the S or RBD structure (van Doremalen et al., Nature, 2020: 1-8; Mercado et al., Nature, 2020: 1-11; Dai et al., Cell, 2020).

Furthermore, several recent studies that have performed neutralization assays against SARS-CoV-2 have demonstrated that the RBD-specific IgG titer and viral neutralization titer have a strong positive correlation. For example, in a study from Li et al, the correlation between the SARS-CoV-2 viral neutralization titer and the RBD-specific IgG titer was 0.622 ($P=0.03$, Li et al., JAMA, 2020, DOI: 10.1001/jama.2020.10044). Suthar et al. reached a similar conclusion in a cohort of 44 PCR-confirmed COVID-19 cases ($r=0.84$, $P < 0.0001$, Suthar et al., Cell Reports Medicine, 2020, DOI: <https://doi.org/10.1016/j.xcrm.2020.100040>). Premkumar et al. collected human sera from 63 SARS-CoV-2 patients and demonstrated that the presence of antibodies targeting the RBD domain is significantly correlated with the abundance of neutralizing antibodies against

SARS-CoV-2 ($r = 0.86$, $P < 0.0001$, Premkumar et. al., Science Immunology, 2020). Therefore, the levels of S- and RBD-specific antibodies in this study could reflect the neutralizing/protective antibody levels to a great degree, making it possible to decipher the dynamic changes in the immune response during COVID-19 infection and recovery. Several conclusions in this study were drawn based on these previous studies. For instances, we found that a fair number of patients were discharged with relatively low levels of S- and RBD-specific IgG. Thus, we reasonably speculated that these patients may have a higher risk of re-infection due to the lack of protective antibodies. We have added the description to the revised Introduction (**Lines 70-91**) accordingly, as well as provided a more accurate statement in the revised manuscript.

Moreover, in order to elucidate the association between virus clearance and antibody response, we further compared the time of viral shedding in patients with different levels of antibody (**Response Figure 2**). Results showed that the patients with weak antibody response had a significantly longer duration of viral shedding than the group with stronger antibody response. This trend was observed in both mild/moderate and severe/critical patients (**Response Figure 3**). Although three kind of antibodies all showed correlations with the duration of viral shedding, this result should be interpreted with caution. It does not indicate that these three antibodies all have the effect of neutralizing the virus, because the generation of these antibodies may be accompanying. We analyzed the correlation among the levels of S-IgG, RBD-IgG, and N-IgG (**Response Figure 4A**). The S-IgG and RBD-IgG levels were highly positive-related because the RBD is part of the S1 unit of the S protein, while the S-IgG/RBD-IgG and N-IgG levels were less strongly correlated. Furthermore, the levels of these three IgG changed jointly in most patients, as illustrated by Patients #508 and #1646 (**Response Figure 4B**). By observing the dynamic changes of viral RNA load and antibody levels in patients with consecutive results we found that there was an association between viral shedding and antibody levels (**Response Figure 5**). For example, in Patient #1086, viral shedding was accompanied by an increase of antibody levels, while Patient #1106 became SARS-CoV-2 RNA negative after the level of S-IgG and RBD-IgG increased, despite a lack

of increase in N-IgG. This part was added to the Result section of the revised manuscript (Lines 238-261).

Response Figure 2 (Related to Figure 2 in revised manuscript). Kaplan–Meier analysis of the viral shedding time in strong responders and weak responders. The y-axis represents the duration of viral shedding (days). The y-axis represents the positive viral RNA rate.

Response Figure 3 (Related to Supplementary Figure S5 in revised manuscript). Kaplan–Meier analysis of the viral shedding time in mild/moderate and severe/critical patients from the strong and weak response groups. The y-axis represents the duration of viral shedding (days). The y-axis represents the positive viral RNA rate.

Response Figure 4 (Related to Figure 3 in revised manuscript). Correlations among S-, RBD-, and N-IgG levels. (A) Scatter plots of the pair-wise correlations among S-, RBD-, and N-IgG levels. Each point represents the IgG level from one sample. The density of points is indicated by colors. (B) Examples of the dynamic changes in S-, RBD-, and N-IgG levels.

Response Figure 5 (Related to Figure 2 in revised manuscript). Dynamic changes in antibody levels and virus RNA load in patients #1086 and #1106. The X-axis represents the detection date. The Y-axis on the left represents the antibody level. The Y-axis on the right represents the CT value from qPCR for the detection of viral RNA load. CT value less than 40 indicated positive SARS-CoV-2 viral infection. Blue dots represent IgG levels, and purple dots represent IgM levels. The *ORF1ab* and *N* genes of SARS-CoV-2 are represented as pink and orange dots, respectively.

6. In Figure 3 serum antibody levels are measured after convalescent plasma transfer (CPT). However, it is impossible to determine whether the CPT had any effect on the measured titers, or on the outcome of their infection, yet the authors conclude that “these observations demonstrate that CPT therapy may provide the long-term antibody against the SARS-Cov-2 virus for patients and help the COVID-19 patients recover”.

Response: Thank you for pointing this out. In order to address this concern, we analyzed the dynamic changes in antibody levels from 1 to 3 days, 4 to 5 days, 6 to 7 days, and 8 to 14 days after CPT in eight patients who received consecutive antibody detection both before and after CPT (**Response Figure 6**). Results showed that the level of RBD-specific and S-specific IgG increased in 7/8 patients within three days after CPT, but the level of N-specific IgG showed no obvious change. Five out of the seven patients who were virus-positive before CPT showed virus-free status or radiologic improvements within 3 days after CPT, indicating the potential role of CPT in providing immediate immunity for recipients. Some retrospective

studies or clinical trials reported that CPT therapy could provide an improvement in the symptoms and a reduction in mortality among patients with severe/critical COVID-19 disease (Li et. al, JAMA, doi: 10.1001/jama.2020.10044; Liu, et al. medRxiv, 2020; Shen C, et al. Jama, 2020, 323(16): 1582-1589; Duan K, et al. PNAS, 2020,117(17):9490-9496). The increase of anti-SARS-CoV-2 antibody levels is one of the most important reasons for the effectiveness of CPT therapy. We added this part to the revised manuscript (**Lines 266-278**):

“We analyzed the changes of levels of S-, RBD-, and N- specific IgG in 1 to 3 days, 4 to 5 days, 6 to 7 days, and 8 to 14 days after CPT in eight patients who received consecutive antibody detection both before and after CPT (**Figure 4**). The level of RBD-specific and S-specific IgG increased in seven of the eight patients within three days after CPT, but the level of N-specific IgG had no obvious change. Five out of seven patients who were SARS-CoV-2 RNA-positive before CPT were RNA negative or showed signs of radiologic improvement within 3 days after CPT, indicating the potential role of CPT in providing immediate immunity for recipients. Some retrospective studies or clinical trials reported that CPT therapy could provide an improvement in the symptoms and a reduction in mortality among patients with severe/critical COVID-19 disease^{15,25,26}. The increase of anti-SARS-CoV-2 antibody levels is one of the most important reasons for the effectiveness of CPT therapy.”

Response Figure 6 (Figure 4 in revised manuscript). Dynamic changes in the S-specific, RBD-specific and N-specific IgG levels after convalescent plasma transfusion therapy in eight patients. (A-C) The x-axis represents the days after convalescent plasma transfusion therapy, and the y-axis represents the antibody level. Different colored lines represent the dynamic changes in antibody levels in different patients. (D) The clinical assessment of patients after receiving COVID-19 convalescent plasma transfusion therapy. Arrows represent the discharge date, and triangles represent radiological improvements. Yellow and green dots represent the date on which the viral RNA tests showed positive and negative detection, respectively.

7. In the aged population the antibodies against the N protein are higher than the anti-N antibodies in the other age groups but the RBD specific levels are lower. The authors attribute this to “a weak immune defense response.” The data don’t support this statement this age cohort is clearly able to mount a robust antibody response, but it’s just focused on the N protein.

Response: Thank you pointing this out. We apologize for not making this clear. It has been recognized that S- or RBD- specific antibodies are neutralizing (Jiang et. al, Trends in Immunology, 2020; Cao, Nat Rev Immunol, 2020), while there is no evidence that N-specific antibodies can block virus infection. It is reported that a large amount of virus load in early infection may lead to higher N-specific antibody levels because of the high immunogenic activity of N protein (Sun et al., Emerging Microbes & Infections, 2020 (just-accepted): 1-36).

Considering that the antibody response against N protein is also part of the immune response, although the antibody is not neutralizing, we revised the manuscript accordingly as follows (Lines 188-199):

“We compared the antibody levels among patients of different age groups (<40 years, 40–65 years, and >65 years) (Table 2). On admission, the RBD-specific antibody level in patients aged >65 years was relatively lower than in younger and middle-aged patients, while the N-specific antibody level of older patients was higher. It has been reported that a high viral load in early infection may cause higher N-specific antibody levels because of the high immunogenic activity of N protein²², but there was no evidence that N-specific antibodies can block viral replication. The S-, RBD-, and N-specific IgG levels were gradually elevated along with the age increase during hospitalization and on discharge ($P<0.05$). For example, the median level of RBD-specific antibody in younger, middle-aged and older patients was 7.7 AU/mL, 22.4 AU/mL, and 30.7 AU/mL, respectively (younger vs. middle-aged: $P<0.001$, middle-aged vs. older: $P=0.003$) during hospitalization.”

Response References:

1. Long Q X, Liu B Z, Deng H J, et al. Antibody responses to SARS-CoV-2 in patients with COVID-19. *Nature Medicine*, 2020: 1-4.
2. Pollán M, Pérez-Gómez B, Pastor-Barriuso R, et al. Prevalence of SARS-CoV-2 in Spain (ENE-COVID): a nationwide, population-based seroepidemiological study. *The Lancet*, 2020.
3. Lerner A M, Eisinger R W, Lowy D R, et al. The COVID-19 Serology Studies Workshop: Recommendations and Challenges. *Immunity*, 2020.
4. To K K W, Tsang O T Y, Leung W S, et al. Temporal profiles of viral load in posterior oropharyngeal saliva samples and serum antibody responses during infection by SARS-CoV-2: an observational cohort study. *The Lancet Infectious Diseases*, 2020.
5. Juno J A, Tan H X, Lee W S, et al. Humoral and circulating follicular helper T cell responses in recovered patients with COVID-19. *Nature Medicine*, 2020: 1-7.
6. Xu X, Sun J, Nie S, et al. Seroprevalence of immunoglobulin M and G antibodies against SARS-CoV-2 in China. *Nature Medicine*, 2020: 1-3.
7. Jiang S, Hillyer C, Du L. Neutralizing antibodies against SARS-CoV-2 and other human coronaviruses. *Trends in Immunology*, 2020.
8. Cao X. COVID-19: immunopathology and its implications for therapy. *Nature Reviews Immunology*, 2020, 20(5): 269-270.
9. Du L, He Y, Zhou Y, et al. The spike protein of SARS-CoV—a target for vaccine and therapeutic development. *Nature Reviews Microbiology*, 2009, 7(3): 226-236.

10. Wu A, Peng Y, Huang B, et al. Genome composition and divergence of the novel coronavirus (2019-nCoV) originating in China. *Cell Host & Microbe*, 2020.
11. Chi X, Yan R, Zhang J, et al. A neutralizing human antibody binds to the N-terminal domain of the Spike protein of SARS-CoV-2[. *Science*, 2020.
12. van Doremalen N, Lambe T, Spencer A, et al. ChAdOx1 nCoV-19 vaccine prevents SARS-CoV-2 pneumonia in rhesus macaques. *Nature*, 2020: 1-8.
13. Mercado N B, Zahn R, Wegmann F, et al. Single-shot Ad26 vaccine protects against SARS-CoV-2 in rhesus macaques. *Nature*, 2020: 1-11.
14. Dai L, Zheng T, Xu K, et al. A universal design of betacoronavirus vaccines against COVID-19, MERS and SARS. *Cell*, 2020.
15. Li L, Zhang W, Hu Y, et al. Effect of Convalescent Plasma Therapy on Time to Clinical Improvement in Patients with Severe and Life-threatening COVID-19: A Randomized Clinical Trial. *JAMA*, 2020.
16. Liu S T H, Lin H M, Baine I, et al. Convalescent plasma treatment of severe COVID-19: A matched control study[J]. *medRxiv*, 2020.
17. Shen C, Wang Z, Zhao F, et al. Treatment of 5 critically ill patients with COVID-19 with convalescent plasma[J]. *Jama*, 2020, 323(16): 1582-1589.
18. Duan K, Liu B, Li C, et al. Effectiveness of convalescent plasma therapy in severe COVID-19 patients[J]. *Proceedings of the National Academy of Sciences*, 2020, 117(17): 9490-9496.
19. Suthar M S, Zimmerman M G, Kauffman R C, et al. Rapid generation of neutralizing antibody responses in COVID-19 patients. *Cell Reports Medicine*, 2020.
20. Liang F Y, Lin L C, Ying T H, et al. Immunoreactivity characterisation of the three structural regions of the human coronavirus OC43 nucleocapsid protein by Western blot: Implications for the diagnosis of coronavirus infection. *Journal of Virological Methods*, 2013, 187(2): 413-420.
21. Leung D T M, Chi Hang T F, Chun Hung M, et al. Antibody response of patients with severe acute respiratory syndrome (SARS) targets the viral nucleocapsid. *The Journal of Infectious Diseases*, 2004, 190(2): 379-386.
22. Sun B, Feng Y, Mo X, et al. Kinetics of SARS-CoV-2 specific IgM and IgG responses in COVID-19 patients. *Emerging Microbes & Infections*, 2020 (just-accepted): 1-36.

Reviewer #2

In this study, Li et al reports a comprehensive study on the kinetics of total antibody, spike protein (S)-, receptor-binding domain (RBD)-, and nucleoprotein (N)- specific IgM and IgG levels during SARS-CoV-2 infection based on samples from 1,850 patients. They found that the S-, RBD-, and N- specific IgG generation of severe/critical COVID-19 patients is one week later than mild/moderate cases, while the levels of these antibodies are 1.5-fold higher in severe/critical patients during hospitalization ($P < 0.01$). The decrease of these IgG levels indicates the poor outcome of severe/critical patients. Notably, they found that the patients who got re-infected had a low level of protective antibody on discharge. Together with previous studies, these findings inform the diagnosis, treatment, as well as vaccine development for COVID-19. However, the concerns below should be addressed by the authors.

1. Please introduce the criteria of patient recruitment. Additionally, how the patients were selected for this study as many patients were accepted for hospitalization in this hospital.

Response: Thank you for the comment. The Wuhan Huoshenshan Hospital is one of the largest designated hospitals for COVID-19 in Wuhan. This hospital was established in early February specifically for COVID-19 treatment and recruited well-trained clinicians and equipped with up-to-date laboratory devices. The detailed laboratory observations make it possible to thoroughly analyze the dynamic changes in the antibodies against SARS-CoV-2 during the infection and recovery of COVID-19. In this study, all enrolled patients admitted from February 4 to March 30 were confirmed to be infected with SARS-CoV-2 using RT-PCR assays on nasopharyngeal swabs. A total of 1850 patients in whom antibodies were detected and with a clear record of symptom onset history were included in this study. We apologize for not making this point clear in the manuscript. We added the following description to the revised manuscript (**Lines 376-386**):

“We analyzed the laboratory test results of COVID-19 patients admitted from February 4 to March 30 to Wuhan Huoshenshan Hospital in China. In this study, all enrolled patients were confirmed to be infected with SARS-CoV-2 using RT-PCR assays performed on nasopharyngeal swab samples. A total of 1850 patients who had been detected antibody levels

and a clear record of symptom onset history were included (Supplementary Table 2). This cohort included 795 patients with mild or moderate disease and 1055 patients with severe or critical disease. Data on the clinical characteristics and laboratory findings of all patients were extracted from the hospital electronic medical records (Supplementary Table 1). This study was approved by the Medical Ethical Committee of Wuhan Huoshenshan Hospital. Written informed consent was obtained from each patient.”

2.How many samples were taken from each patient? How the consecutive samples were collected? How about the interval of sampling?

Response: Thank you for pointing this out. Laboratory tests of 1850 hospitalized COVID-19 patients were analyzed (**Supplementary Table 1**). For the detection of total IgM/IgG, 3063 serum samples from 1850 patients were tested. Each patient was tested 1 to 10 times, and 673 (36.4%) were tested more than once. The median sampling interval was 5 days among patients who were tested more than once. For the detection of S-, RBD- and N- specific antibodies, 710 serum samples from 418 patients were tested. Each patient was tested 1 to 7 times, and 169 (40.4%) were tested more than once. The median sampling interval was 4 days among patients who were tested more than once. We added the following description to the revised manuscript (**Lines 107-114**):

“For the detection of total IgM/IgG, 3063 serum samples from 1850 patients were tested. Each patient was tested 1 to 10 times, and 673 (36.4%) were tested more than once. The median sampling interval was 5 days among patients who were tested more than once. For the detection of S-, RBD- and N- specific antibodies, 710 serum samples from 418 patients were tested. Each patient was tested 1 to 7 times, and 169 (40.4%) were tested more than once. The median sampling interval was 4 days among patients who were tested more than once.”

3.The most interesting data related to the patient who was re-infected after discharge. Because the duration between the two hospitalization is very close, how to exclude the persistent infection? As some patients were found to shed virus in a long term.

Response: We agree with the reviewer that the false-negative results of viral RNA tests of nasopharyngeal swabs maybe one possible reason for re-detectable positive RNA test among recovered COVID-19 patients. In contrast, we did observe that a considerable number of patients were discharged with a low level of antibodies. Specifically, in the patients who received antibody detection within 3 days before discharge, 10.94% (72/658) of the patients were negative for total IgG, and the negative rate of S-IgG, RBD-IgG and N-IgG were 5.10% (10/196), 7.14% (14/196) and 8.67% (17/196), respectively. Given the important role of S-IgG and RBD-IgG in neutralizing SARS-CoV-2, it is reasonable to speculate that the patients who discharged without these protective antibodies may have a high risk of re-infection. Our observation suggested that patients with negative throat or nasopharyngeal swabs but low antibody levels need to raise discharge criterion, or strengthen protection and monitoring after discharge. We added description in Results (**Lines 281-299**) and discussion (**Lines 336-356**) sections to make this clear as:

“Moreover, there have been reports that some patients have re-detectable positive RNA tests after recovery³⁰, raising questions about the patients’ condition on discharge. According to the latest version of "Diagnosis and Treatment Protocol for Novel Coronavirus Pneumonia of China", patients who meet the following criteria can be discharged: (1) a return of body temperature to normal for more than three days; (2) an obvious improvement in respiratory symptoms; (3) pulmonary imaging shows obvious absorption of inflammation; and (4) two consecutive negative nucleic acid tests on respiratory tract samples such as sputum and nasopharyngeal swabs (measured on samples collected at least 24 hours apart). However, our study shows that patients with re-detectable positive RNA test results tended to have low antibody levels on discharge, highlighting the importance of antibody testing prior to discharge. In our opinion, there are two possible reasons for re-detectable positive RNA test results among recovered COVID-19 patients. First, studies have shown that SARS-CoV-2 mainly infects the lower respiratory tract³¹, but the collection of the bronchoalveolar lavage requires skilled operators and specific devices, exposing the clinicians to a high risk of infection. So, the nasopharyngeal swab samples are used to assess the viral load, the sample quality may lead to false-negative results. Second, the immune defense system of some discharged patients was relatively weak. Lacking sufficient antibody may make patients susceptible to re-infection. Thus, patients with negative nasopharyngeal swabs but low S-specific- or RBD-specific- IgG levels need to raise discharge criterion, or strengthen protection and monitoring after discharge.”

4. The authors should discuss why the antibody titer was lower in the early weeks in severe cases and obviously increased then? But the titers decreased sharply in the non-survivors?

Response: Thank you for your comments. The antibody level was observed to be lower in the early weeks in severe cases. This may be because the antibody response was late among these patients, as we observed that the profiles of antibody changes were various in different patients. We added the corresponding discussion in the revised manuscript.

Because only 5 non-survivors received the detection of S-, RBD-, and N- specific antibodies, and lacked enough consecutive observations, it is hard to tell if the trend is real. However, the total IgM and IgG levels were tested in 46 serum samples from 21 non-survivors. By calculating the average IgG/IgM level of each patient, we compared the average antibody levels between survivors and non-survivors. Results showed that the antibody levels in non-survivors were significantly lower than those in survivors (**Response Figure 7**). We added this to the revised manuscript (**Lines 204-212**) as:

“The total IgM and IgG levels were tested in 46 serum samples from 21 non-survivors. We compared the average antibody levels between survivors and non-survivors by calculating the average IgG/IgM level of each patient. The antibody levels in non-survivors were significantly lower than those in survivors ($P=0.04$ and $P=0.03$ for IgM and IgG, respectively, **Supplementary Figure S4**), suggesting that the antibody response played important roles in helping the severe/critical COVID-19 patients recover. It was not possible to compare the S-, RBD-, and N-specific antibody levels in non-survivors and survivors due to a lack of data in non-survivors.”

Response Figure 7 (Supplementary Figure S4 in revised manuscript). Total IGM and IGG levels in survivors and non-survivors of COVID-19 patients.

5. The relationship between viral shedding and antibody can be discussed.

Response: Thank you for your suggestion. In order to analyze the association between virus clearance and antibody response, we compared the time of viral shedding in patients with different levels of antibody (**Response Figure 8**). Results showed that patients with weak antibody response had a significantly longer duration of viral shedding than patients with a stronger antibody response. This trend was observed in both mild/moderate and severe/critical patients (**Response Figure 9**). Although three kind of antibodies all showed correlations with the virus shedding duration, this result should be interpreted cautiously. It does not indicate that these three antibodies all have the effect of neutralizing the virus, because the generation of these antibodies may be accompanying (**Response Figure 10**). We illustrated the dynamic changes of both viral RNA load and antibody levels in the same set of patients who received consecutive detection (**Response Figure 11**). It is shown that the trend of viral shedding was associated with the antibody levels. We added this analysis in the revised manuscript (**Lines 238-261**) as:

“We compared the time of viral shedding in patients with different levels of antibodies. The maximum level of S-, RBD-, and N-specific antibodies of each patient was calculated, and the top 33% and bottom 33% patients were categorized as strong-responders and weak-responders. The duration of virus shedding was defined from the symptom onset to the date of the last positive nasopharyngeal swab. Patients with a weak antibody response had a significantly longer duration of viral shedding than those with a strong antibody response (Log rank $P=0.019$, 0.014 , and <0.001 for S-, RBD-, and N- specific IgG, respectively, **Figure 2C**). This trend was observed in patients with both mild/moderate and severe/critical disease (**Supplementary Figure S5**). Although three kind of antibodies all showed correlations with the duration of viral shedding, this result should be interpreted with caution. It does not indicate that these three antibodies all have the effect of neutralizing the virus, because the generation of these antibodies may be accompanying. We analyzed the correlation among the levels of S-IgG, RBD-IgG, and N-IgG (**Figure 3A**). The S-IgG and RBD-IgG levels were highly positive-related because the RBD is part of the S1 unit of the S protein, while the S-IgG/RBD-IgG and N-IgG levels were less strongly correlated ($R=0.504$

between S-IgG and N-IgG, and $R=0.397$ between RBD-IgG and N-IgG). Furthermore, the levels of these three IgG changed jointly in most patients, as illustrated by Patients #508 and #1646 (Figure 3B). By observing the dynamic changes of viral RNA load and antibody levels in patients with consecutive results we found that there was an association between viral shedding and antibody levels (Figure 2D and Supplementary Figure S6). For example, in Patient #1086, viral shedding was accompanied by an increase of antibody levels, while Patient #1106 became SARS-CoV-2 RNA negative after the level of S-IgG and RBD-IgG increased, despite a lack of increase in N-IgG.”

Response Figure 8 (Related to Figure 2 in revised manuscript). Kaplan–Meier analysis of the viral shedding time in patients of the strong-response and weak-response groups. The y-axis represents the duration of viral shedding (days). The x-axis represents the positive rate of viral RNA.

Response Figure 9 (Supplementary Figure S5 in revised manuscript). Kaplan–Meier analysis of the viral shedding time in mild/moderate or severe/critical patients of the strong-response and weak-response groups. The y-axis represents the duration of viral shedding (days). The y-axis represents the positive rate of viral RNA.

Response Figure 10 (Figure 3 in revised manuscript). Correlations among S-, RBD-, and N- IgG levels. (A) Scatter plots of the pair-wise correlations among S-, RBD-, and N- IgG levels. Each point represents the IgG level of one sample. The density of points was shown using different color. (B) Examples for the dynamic changes of S-, RBD-, and N- IgG levels.

Response Figure 11 (Related to Figure 2 in revised manuscript). The dynamic changes of antibody levels and virus RNA load in patients #1086 and #1106. The X-axis represents the detection date. The Y-axis on the left represents antibody level. The Y-axis on the right represents the CT value of PCR for the detection of viral RNA load. CT value less than 40 is defined as SARS-CoV-2 viral positive. Blue dots represent IgG levels and the purple dots represent IgM levels. The *ORF1ab* and *N* genes of SARS-CoV-2 were represented as pink and orange dots respectively.

6. The sample size should be marked in legends of figure 2 and 3.

Response: Thank you for your suggestion. We have added the number of samples in the revised figures where appropriate, including **Figure 1, Figure 2, Figure 5, and Supplementary Figure S3, S4, S5.**

7. The demographic, clinical and laboratory data of the patients recruited should be disclosed by a table.

Response: Thank you for your suggestion. We have added the demographic, clinical and laboratory data in the revised **Supplementary Table S1** as follows.

	Mild/Moderate (N=795)	Severe/Critical (N=1055)
Age (yr.)– median (IQR)	47 (57-65)	55 (64-71.5)
Sex– no. (%)		
Female	396 (49.8)	525 (49.8)
Male	399 (50.2)	530 (50.2)
Comorbidity – no. (%)		
Hypertension	217 (27.3)	389 (36.9)

Diabetes	107 (13.5)	181 (17.2)
Cardiovascular disease	68 (8.6)	164 (15.5)
Cerebrovascular disease	26 (3.3)	72 (6.8)
Malignancy	15 (1.9)	42 (4)
Chronic obstructive pulmonary disease	28 (3.5)	79 (7.5)
Chronic renal disease	8 (1.0)	23 (2.2)
Chronic liver disease	25 (3.1)	30 (2.8)
Immunodeficiency	2 (0.3)	5 (0.5)
Days from symptoms onset to admission(d) – median (IQR)	33 (22-38)	30 (15.8-35)
Days from admission to discharge(d) – median (IQR)	11 (7-15)	14 (9-24)
ICU admission– no. (%)	0 (0)	63 (6.0)
Clinical outcomes – no. (%)		
Discharge from hospital	790 (99.4)	1004 (95.2)
Death	0 (0)	21 (2.0)
Hospitalization	5 (0.6)	30 (2.8)
Laboratory findings–median (IQR)		
LYM (%)	29 (23.1-34.3)	23.3 (13.4-30.9)
Mono (%)	7.4 (6.2-8.7)	7.4 (6-9.1)
NEUT (%)	60 (54.1-66.1)	65.2 (56.8-76.8)
LDH (IU/L)	163.4 (144.6-187.1)	201.6 (169.2-252.5)
BNP (pg/mL)	0.01 (0.01-13.59)	28.76 (0.01-130.96)
CRP (mg/L)	1.4 (0.6-3.4)	3.3 (1-15.3)
Urea (mmol/L)	4.5 (3.7-5.5)	5 (4-6.7)
PCT (ng/ml)	0.04 (0.03-0.05)	0.06 (0.04-0.12)
Fdg (g/L)	2.9 (2.6-3.2)	3 (2.6-3.5)
hs-cTnl(ng/ml)	0.01 (0.01-0.01)	0.01 (0.01-0.01)
GLU (mmol/L)	4.8 (4.5-5.5)	5.4 (4.7-7.2)

8.The limitation of the study should be presented in the discussion section.

Response: Thank you for your advice. We added the limitation in the discussion section of revised manuscript (**Lines 356-368**) as:

“There are some limitations in this study. First, just as the cumulative observations shown, the false negative of viral RNA test of nasopharyngeal swabs may be inevitably included in this study. Second, as one of the largest designated hospitals which was built specifically for COVID-19 pandemic in Wuhan, Huoshenshan Hospital admitted some patients transferred from other non-designated hospitals when it opened in early February; hence, it is difficult to reconstruct the temporal development of antibodies in these patients. Whether the antibodies against S, RBD, and N proteins develop at the same time or sequentially still needs further prospective studies. Third, ideally, this study should be based on continuous time point detection, but it was not possible to conduct continuous systematic sample collection during the pandemic, and not all patients had continuous observation data, limiting the power of consecutively observing the antibody response in certain individuals.”

9.The median time of antibody occurrence should be compared with other studies.

Response: Thank you for your professional advice. The ideal way to observe the timing of the antibody occurrence is to test the antibody levels of close contacts before or at the very beginning of symptom onset. However, Wuhan Huoshenshan Hospital was built for specifically treating COVID-19 patients in early February. At that time, due to the sudden outbreak of the pandemic, a number of people could not go to the designated hospital. As one of the largest designated hospitals for COVID-19 in Wuhan, Huoshenshan hospital has admitted some patients transferred from other non-designated hospitals in early February. Therefore, it is difficult to capture the antibody generation timing for these patients. Of all the patients who had clear record of symptom onset history, only 48 of them received antibody detection within the first week after symptom onset.

The dynamic laboratory observations in a large cohort make it possible for us to analyze the positive rate of IgG/IgM from the first to 12 weeks after symptom onset. The previous knowledge learned from SARS and MERS shows that the IgM antibody produces within 3-6 days after symptoms onset, and the production of IgG antibody lags behind the IgM, which is detectable after 8 days of symptom onset (Li et al., Journal of medical virology, 2020). In our study of COVID-19, we observed that IgG and IgM can be detectable in 70.83% and 39.58% of patients, respectively, within the first week of onset (**Response Figure 12**), which indicates that IgG shows rapid response during the early infection of SARS-CoV-2. Consistent with our observation, in the study from Long et al., 31.8% and 13.6% of patients generated IgG and IgM, respectively, within 2-4 days, and 55.6% and 40% of patients generated IgG and IgM, respectively, within 5-7 days (Long et. al, Nature Medicine, 2020). A study by Xu et al. mentioned that it was surprising to see an early and a higher level of IgG, suggesting that the value of IgM as an early marker for the acute phase of SARS-COV-2 infection might not be on par with that in other viral infection diagnostics (Xu et al., Nature Medicine, 2020). In our opinion, the observed high positive rate of IgG at the early stage of SARS-CoV-2 infection may be because some COVID-19 patients are asymptomatic at the beginning of disease onset (Long et. al, Nature Medicine, 2020). According to a recent report, 97.5% of people who develop symptoms do so within 11.5 days (Wiersinga et al., JAMA, 2020). We discussed this in the revised manuscript (**Lines 138-148**) as:

“We further quantified the total antibody levels of patients with confirmed SARS-CoV-2 infection, and found that 39.6% of patients were IgM positive, and 70.8% were IgG positive within the first week after symptom onset (**Figure 1C**). IgG could be detected in 95.3% of patients 5 weeks after symptom onset. Consistent with our observations, previous studies also found that the positive rate of IgG was higher than IgM, unlike the previous experiences from some other infectious diseases including SARS-CoV-1¹⁹. Xu et al.¹⁸ mentioned that it was surprising to see an early and a higher level of IgG, suggesting that the value of IgM as an early marker for the acute phase of SARS-COV-2 infection might not be on par with that in other viral infection diagnostics. This phenomenon may result from that some COVID-19 patients are asymptomatic at the beginning of infection^{20,21}.”

Response Figure 12 (Related to Figure 1 in revised manuscript). The positive rate of total IgM and IgG level. The x-axis indicates the weeks after symptom onset, and the y-axis shows the antibody-positive rate among confirmed COVID-19 patients.

Response References:

1. Li Z, Yi Y, Luo X, et al. Development and clinical application of a rapid IgM-IgG combined antibody test for SARS-CoV-2 infection diagnosis. *Journal of Medical Virology*, 2020.
2. Long Q X, Liu B Z, Deng H J, et al. Antibody responses to SARS-CoV-2 in patients with COVID-19. *Nature Medicine*, 2020: 1-4.
3. Xu X, Sun J, Nie S, et al. Seroprevalence of immunoglobulin M and G antibodies against SARS-CoV-2 in China. *Nature Medicine*, 2020: 1-3.
4. Long Q X, Tang X J, Shi Q L, et al. Clinical and immunological assessment of asymptomatic SARS-CoV-2 infections. *Nature Medicine*, 2020: 1-5.
5. Wiersinga W J, Rhodes A, Cheng A C, et al. Pathophysiology, Transmission, Diagnosis, and Treatment of Coronavirus Disease 2019 (COVID-19): A Review. *JAMA*.2020.

REVIEWERS' COMMENTS

Reviewer #1 (Remarks to the Author):

The revised manuscript has adequately addressed the issues I had with the initial submission.

Reviewer #2 (Remarks to the Author):

In this study, Li et al report dynamic changes in anti-SARS-CoV-2 antibodies during SARS-CoV-2 infection and recovery from COVID-19. However, a number of studies has described the characteristics of humoral response after SARS-CoV-2 infection over the past months. The novelty is not very strong at this time. The data on the recovery patients is novel. Most of the concerns have been clarified except the two points below:

- (1)The authors did not disclose which guideline or standard was used for diagnosing patients.
- (2)The Authors said that "each patient was tested 1 to 10 times", please show how mam samples were taken from each patient?

Response

Reviewer #1 (Remarks to the Author):

The revised manuscript has adequately addressed the issues I had with the initial submission.

Response: We thank the reviewer for reviewing our work again. We are delighted that all concerns have been successfully addressed.

Reviewer #2 (Remarks to the Author):

In this study, Li et al report dynamic changes in anti-SARS-CoV-2 antibodies during SARS-CoV-2 infection and recovery from COVID-19. However, a number of studies has described the characteristics of humoral response after SARS-CoV-2 infection over the past months. The novelty is not very strong at this time. The data on the recovery patients is novel. Most of the concerns have been clarified except the two points below:

(1) The authors did not disclose which guideline or standard was used for diagnosing patients.

Response: We thank the reviewer for reviewing our work again. The diagnosis of patients in Wuhan Huoshenshan Hospital was according to the Diagnosis and Treatment Protocol for Novel Coronavirus Pneumonia released by the National Health Commission (Version 6 or trail version 7) in China. According to this guideline, patients were diagnosed using viral RT-PCR detection of nasopharyngeal swab samples. We have revised the text as follows: “We analyzed the laboratory test results of COVID-19 patients admitted from February 4 to March 30 to Wuhan Huoshenshan Hospital in China, in which patients were diagnosed based on the Diagnosis and Treatment Protocol for Novel Coronavirus Pneumonia released by the National Health Commission (Version 6 or trail version 7, <https://www.who.int/docs/default-source/wpro---documents/countries/china/covid-19-briefing-nhc/1-clinical-protocols-for-the-diagnosis-and-treatment-of-covid-19-v7.pdf>). In this study, all enrolled patients were confirmed to be infected with SARS-CoV-2 using RT-PCR assays performed on nasopharyngeal swab samples (Supplementary Table 2). A total of 1850 patients who had been detected antibody levels and a clear record of symptom onset history were included (Supplementary Data 2).”

(2) The Authors said that “each patient was tested 1 to 10 times”, please show how many samples were taken from each patient?

Response: Thank you for your comments. We have added the number of samples of each patient in Supplementary Data 1. Moreover, the raw data of antibody levels of each sample were shown in Supplementary Data 2.